# A genomic catalogue of soil microbiomes boosts mining of biodiversity and genetic resources

Bin Ma [1,2,3,7], Caiyu Lu[1,2,3,7], Yiling Wang [1,2,3,7], Jingwen Yu[3], Kankan Zhao [1,2], Ran Xue[3], Hao Ren[3], Xiaofei Lv[4], Ronghui Pan[3], Jiabao Zhang [5], Yongguan Zhu [6] & Jianming Xu [1,2] ✉

Soil harbors a vast expanse of unidentified microbes, termed as microbial dark matter, presenting an untapped reservoir of microbial biodiversity and genetic resources, but has yet to be fully explored. In this study, we conduct a large-scale excavation of soil microbial dark matter by reconstructing 40,039 metagenome-assembled genome bins (the SMAG catalogue) from 3304 soil metagenomes. We identify 16,530 of 21,077 species-level genome bins (SGBs) as unknown SGBs (uSGBs), which expand archaeal and bacterial diversity across the tree of life. We also illustrate the pivotal role of uSGBs in augmenting soil microbiome's functional landscape and intra-species genome diversity, providing large proportions of the 43,169 biosynthetic gene clusters and 8545 CRISPR-Cas genes. Additionally, we determine that uSGBs contributed 84.6% of previously unexplored viral-host associations from the SMAG catalogue. The SMAG catalogue provides an useful genomic resource for further studies investigating soil microbial biodiversity and genetic resources.

The soil microbiome is not only valuable as the primary regulator of soil ecosystem services but also as a source of genetic resources for human healthcare and biotechnological applications[1]. The majority of antibiotics currently used in human medicine were discovered from soil-living bacteria or fungi between the 1940s and 1970s[2], but the golden age declined after the 1970s owing to the difficulty of cultivating unidentified bacterial species[3]. However, cultivation-independent approaches, e.g., rRNA gene-based survey, have confirmed that up to 99% of soil microorganisms have not been cultivated under laboratory conditions to date[4]. Those countless undiscovered microbes in the soil, referred to as soil microbial dark matter[5], comprise enormous untapped diversity and genetic resources[6]. For

instance, Ling et al. recently discovered teixobactin, a new antibiotic without detectable resistance, by growing uncultured bacteria from the soil with iChip[7].

Genome-resolved metagenomics can yield metagenome-assembled genomes (MAGs) from contigs assembled with shotgun-sequenced short reads[8], providing previously unexplored genomes of Bacteria[9], Archaea[10], and viruses[11] for understanding the functional characteristics of uncultivated microbes. MAGs have substantially expanded the genomic catalogue for manifold environments including relatively limited soil environments[12], the human gut[13], animals[14], the global ocean[15], and other environments[16,17]. In additionally, MAGs have increased the diversity and topological structure of the tree of life,

[1]Institute of Soil and Water Resources and Environmental Science, College of Environmental and Resource Sciences, Zhejiang University, Hangzhou 310058, China. [2]Zhejiang Provincial Key Laboratory of Agricultural Resources and Environment, Zhejiang University, Hangzhou 310058, China. [3]ZJU-Hangzhou Global Scientific and Technological Innovation Center, Hangzhou 311200, China. [4]Department of Environmental Engineering, China Jiliang University, Hangzhou 310018, China. [5]State Key Laboratory of Soil and Sustainable Agriculture, Institute of Soil Science, Chinese Academy of Sciences, Nanjing 210008, China. [6]Research Center for Eco-environmental Sciences, Chinese Academy of Sciences, Beijing 100085, China. [7]These authors contributed equally: Bin Ma, Caiyu Lu, Yiling Wang. ✉e-mail: jmxu@zju.edu.cn

providing insights into uncultivated microbial taxa and virus-host associations, as well as promoting the discovery of genetic resources, such as biosynthetic gene clusters (BGCs)[18], CRISPR[19], and antiphage defense systems[20].

A single gram of surface soil can contain billions of bacterial and archaeal cells and trillions of viruses[21], indicating that soil microbial diversity is substantially higher than in other environments due to its high complexity and spatial heterogeneity[4]. However, few prior studies focused on reconstructing MAGs from soils, due to the challenges associated with complicated soil metagenomes, which are enriched genomes for uncultivated and undescribed microorganisms[1]. Most existing studies on the soil microbiome suffer from the limitations and biases of reference databases and cannot characterize microbes with high taxonomic resolution[22]. Several studies have recovered genomes from soil metagenomes at a small scale and multi-systems for exploring their functions and genetic resources[15,23], but the myriad of soil metagenomes available in public databases has not been mined at present at a global scale.

In this work, to construct an informational public resource database and explore soil microbial dark matter from metagenomes, we first reconstructed MAGs from global-scale genome-resolved metagenomics to expand the genomic catalogue of soil microbiomes and shed light on microbial dark matter in soils. We then clustered the MAGs into 21,077 SGBs and identified 16,530 uSGBs by aligning SGBs with ~500,000 reference genomes from the Refseq database and MAGs from other studies. Intraspecific pangenome and single nucleotide variants (SNVs) profiles reveal the functional contribution of uSGBs in the soil microbiomes. Moreover, we explored BGCs and CRISPR-Cas genetic resources, confirming the considerable potential of soil microbiomes in mining genetic resources. Furthermore, we uncovered previously unexplored viral-host associations concealed in the MAGs. The SMAG catalogue constitutes abundant information, providing important opportunities for future broad studies focused on unraveling the ecological roles of soil microbiomes and identifying genetic resources.

## Results

### 40,039 MAGs reconstructed from large-scale genome-resolved metagenomics

To reconstruct previously unexplored bacterial and archaeal genomes, we performed a large-scale single-sample metagenomic assembly on 3304 soil metagenomes across the globe (Fig. 1a), including 363 metagenomes from the in-house dataset and 2941 from publicly available metagenomes. The soil samples were mainly collected from grassland, cultivated land, and forest (Fig. 1b). The number of reconstructed MAGs per metagenome was positively correlated with metagenome read depth (Supplementary Fig. 1a) and follows a power-law distribution (Supplementary Fig. 1b). The number of reconstructed MAGs substantially increased when the number of clean reads >$10^8$ (Supplementary Fig. 1a), suggesting that sequencing depth greater than this threshold would result in worthwhile gains in MAG reconstruction. We reconstructed a total of 40,039 genomes that meet or exceed the medium-quality level of the minimum information about a metagenome-assembled genome (MIMAG) standard[24] (completeness ≥50% and contamination <10%), which we refer to as the SMAG catalogue (Supplementary Data 2). About 3641 (9.1%) of these MAGs were identified as high-quality genomes with completeness >90%, contamination <5%, and presence of the 23S, 16S, and 5S rRNA gene and at least 18 tRNAs according to recent guidelines (Fig. 1a, Supplementary Fig. 1c–e). Moreover, 5184 (13%) of MAGs had completeness ≥90% and contamination <5%, but the absence of all rRNA genes or less than 18 tRNAs[24], largely meaning that near full-complement rRNA genes sequences are challenging for assembling from metagenomes[25], especially for near-complete MAGs[26]. To evaluate the quality of the MAGs in the SMAG catalogue, we inferred the level of strain heterogeneity within each MAG. The median strain heterogeneity (proportion of polymorphic positions) of the high-quality SMAG catalogue was 7.14% (Supplementary Fig. 1f). And the SMAG catalogue is distinct in its exclusive focus on soil microbiomes on a global scale, which specifically allowed us to undertake an in-depth analysis of this particular niche, expanding the knowledge base on soil microbial diversity. Besides, the geographic distribution of the soil metagenomes in our study significantly substantially extended compared with the MAGs resource from environments (Fig. 1d), surpassing both MAGs derived from the Tara Ocean project[27,28] and environmentally derived MAGs from Genomes of Earth's Microbiomes (GEM) catalogue[29], but lagging behind human-associated MAGs[13,30], highlighting the challenges posed by the complex and heterogeneous soil environment[4] and also illustrating the necessity of constructing high-quality soil metagenomic genome reference datasets for more accurate predictions about the ecological functions of soil microbiomes.

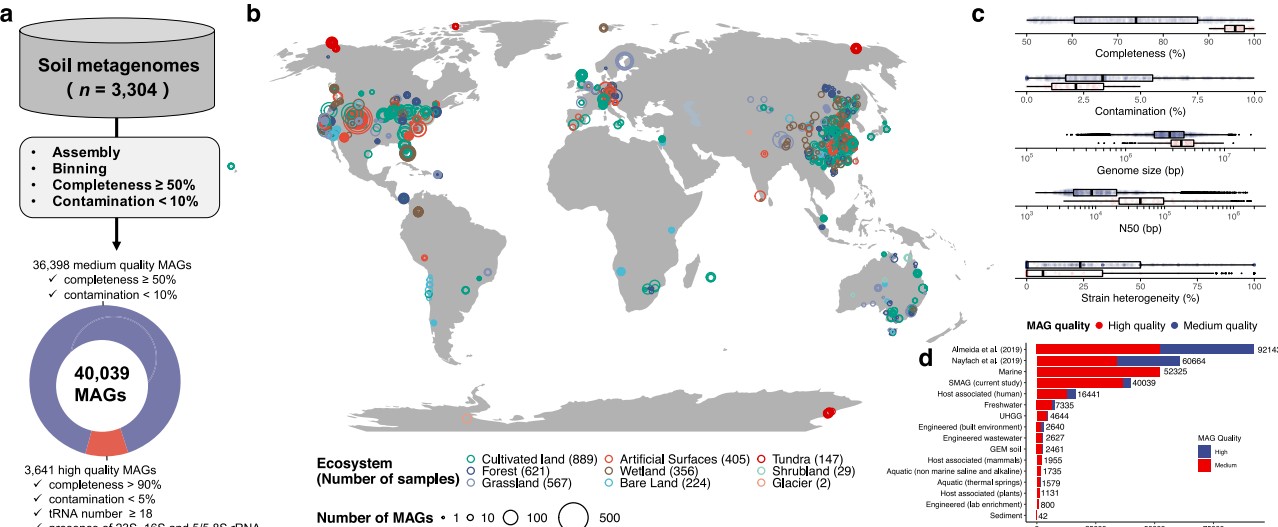

**Fig. 1 | Recovery of genomes from globally distributed soil metagenomes. a** A total of 40,039 MAGs were recovered from 3304 soil metagenomes. **b** Geographic distribution of metagenomes within each habitat. **c** Distribution of quality metrics across the MAGs. **d** Comparison of the current dataset with the published MAG catalogue across different environments; UHGG (Unified Human Gastrointestinal Genome).

## MAG analyses retrieve 16,530 previously uncharacterized bacterial and archaeal clades

To explore taxonomic components in the SMAG catalogue, we clustered the 40,039 MAGs into 21,077 SGBs (Supplementary Data 2) based on 95% whole-genome average nucleotide identity (ANI). We annotated the taxonomy with Genome Taxonomy Database (GTDB), which is commonly used[13] and considered a gold standard for defining prokaryotic species[31] (Fig. 2a). These SGBs were assigned to 88 bacterial phyla and 11 archaeal phyla. The number of MAGs in the SGBs follows a power-law distribution (Supplementary Fig. 2a), suggesting that most of the SGBs comprised a few MAGs.

To identify previously unexplored soil bacterial and archaeal clades in the SMAG catalogue, we compared the MAGs from the SMAG against nearly 500,000 reference genomes, including 282,219 genomes from the Refseq database (November of 2021), 207,953 MAGs from previous studies, and 123,580 MAGs and 1710 single-cell amplified genomes (SAGs) from GenBank (November of 2021). We identified 16,530 uSGBs (78.4% of SGBs) and 4567 known SGBs, (22.6% of SGBs) (Fig. 2a) based on the threshold of 95% ANI and 30% alignment fraction (AF). Consistent with the knowledge of soil microbial dark matter[32], we also found that most MAGs in the SMAG catalogue are uSGBs. The genome size of SGBs and the reference genome size showed a positive linear relationship (Supplementary Fig. 2b). Moreover, we found that most of the SGBs (70.8%) and uSGBs (71.4%) were singleton MAGs (Fig. 2a). The proportion of singleton MAGs in uSGBs (71.2%) was substantially higher than in known SGBs (kSGBs) (50.0%) (Fig. 2a), indicating the critical contribution of the SMAG catalogue in recovering rare species of soil microbiomes. The vast majority of SGBs were unannotated at the species level by the GTDB (18,988, 90.1%), and were barely aligned to reference genomes (14,060, 88.6% of uSGBs with <90% ANI or <10% AF compared to reference genomes).

To examine whether the previously unidentified uSGBs in the SMAG catalogue improve mappability for soil metagenomes, we mapped 494 metagenomes randomly selected from the metagenomes dataset for reconstructing the SMAG catalogue to all 40,039 MAGs. The total mapping rates (the ratio of mapped reads to the total reads) ranged from 2.6 to 89% (medium mapping rate = 12.5%) (Fig. 2b). Consistent with the previous study[33], the contribution of uSGBs for reads mapping was fivefold of kSGBs, which illustrated that the uSGBs were important genomic resources to understand soil microbial dark matter. Moreover, the mapping capacity of the SMAG catalogue was further validated by aligning 70 other soil metagenomes unused for reconstructing the SMAG catalogue (Fig. 2b). The genome size of all recovered MAGs ranged from 0.53 to 12.3 Mb (Supplementary Fig. 2c). Most phyla's genome sizes were consistent between kSGBs and uSGBs, except for Armatimonadota, Bdellovibrionota, and unclassified bacterial phylum UBA10199. MAG sizes of uSGBs were larger than MAG sizes of kSGBs for these phyla. (Supplementary Fig. 2c). MAGs sizes of kSGBs were consistent with the genome size of isolated reference genomes of the same genera (Supplementary Fig. 2b), which validates metagenomic-driven strategies to mine the uSGBs from the complex soil environment. The phyla with the smallest genome sizes are Patescibacteria (median = 0.78 Mb) and Thermoproteota (median = 1.60 Mb), especially for Patescibacteria with large function size by simplifying genome size[34], while Myxococcota (median = 5.10 Mb), Cyanobacteria (median = 5.07 Mb), and Planctomycetota (median = 4.79 Mb) have the largest genome sizes. The smallest-sized high-quality MAGs (~0.53 Mb) were assigned to a previously unidentified species of the genus *Buchnera*, which is experiencing a reductive process towards a minimum genome needed for symbiotic life with aphids[35].

Next, we built the phylogenetic tree of 21,077 SGBs (Fig. 2d), showing that the bacterial and archaeal diversity across the tree of life was expanded by uSGB genomes from the SMAG catalogue. The proportions of uSGBs in the eight most dominant bacterial phyla (>75%)

were greater than those in most of the rare phyla except Planctomycetota, Armatimonadota, and Eremiobacterota (>80%) (Fig. 2e), demonstrating the challenges in assembling rare biosphere[36]. Although the MAGs were assembled from corresponding soil samples as dominant taxa, they were rare in most of the other soil samples. And these MAGs may provide tremendous reference genomic resources in deciphering potential functions of rare biosphere in soil microbiomes (Fig. 2f).

Based on values of relative evolutionary divergence (RED)[37] in the GTDB (release 202) annotation (Supplementary Fig. 2d), we further identified previously unidentified lineages at higher taxonomic ranks. In total, we determined 6392 unannotated genus-level genome bins (uGGBs), 1166 unannotated family-level genome bins (uFGBs), 258 unannotated order-level genome bins (uOGBs), and 31 unannotated class-level genome bins (uCGBs) by GTDB-tk. Two bacterial SGBs were potentially unannotated phylum-level genome bins (uPGBs) with completeness and contamination at (90.65%, 2.44%), and (90.96%, 1.10%), respectively, which indeed illustrated the underestimated diversity of the soil microbial dark matter and highlighted the pressing need for continued exploration of the soil microbiome. This is based on the concatenated protein phylogeny as the basis for a bacterial taxonomy[37], which improved the classification of uncultured microorganisms of the SMAG. However, the rarefaction curves reveal obvious unsaturation at species rank in the SMAG catalogue (Supplementary Fig. 2), indicating that additional previously uncharacterized lineages are yet to be discovered at species ranks.

### Functional landscape and intraspecies genomic diversity

To better understand the functional landscape of soil microbiota, we predicted full-length putative protein sequences from the 5184 high-quality MAGs (>90% completeness, <5% contamination) of the SMAG catalogue. We then performed an in-depth functional annotation of those gene clusters with eggNOG database[38] (v5.0). We identified 41 KEGG pathways, most of which were enriched in uSGBs (Supplementary Fig. 3a), including the pathways related to polyketide synthesis and disease association pathways. Based on the KEGG enrichment analysis, many phyla were only annotated by functional enrichment from uSGBs in the SMAG catalogue, especially for Asgardarchaeota, Krumholzibacteriota, and Tectomicrobia (Fig. 3a). Besides, for the COG functional categories, we also found the Function unknown was over-represented in the SMAG catalogue, and uSGBs in particular (Supplementary Fig. 3b), providing evidence that uSGBs substantially expanded the functional landscape.

Core genes are shared by all strains that are involved in basic biological processes, such as gene expression, energy production, and amino acid metabolism. Accessory genes are the specific genes for certain genomes. To explore the intraspecific genomic diversity of the SMAG, we generated 107 pangenomes for 2200 SGBs with >10 high-quality MAGs by clustering protein sequences from all conspecific genomes at 90% amino acid identity, which was used to define a "core" genome[13]. Open pangenomes have larger sizes with the increase of individuals[39]. To assess the openness of pangenomes of the SMAG, we identified that the longest pangenome size is 20,926,893 bp with 14 conspecific genomes. The average pangenome length reached 6,699,815 bp and almost 40% of the pangenome size is larger than the average. (Supplementary Fig. 3c, Supplementary Data 3). The proportion of core genes decreased with the number of conspecific genomes and genome sizes (Supplementary Fig. 3d, e), which is consistent with previous studies on a limited number of strains and species due to the addition of duplicated genes[40].

The proportion of core genes varied across different phyla (Fig. 3b). Species from Verrucomicrobiota and Nitrospirota showed the highest and the lowest proportion of core genes, respectively (Fig. 3b), which is mainly due to their species ubiquity[41]. Given that Verrucomicrobia is generally among the most abundant taxa in soil

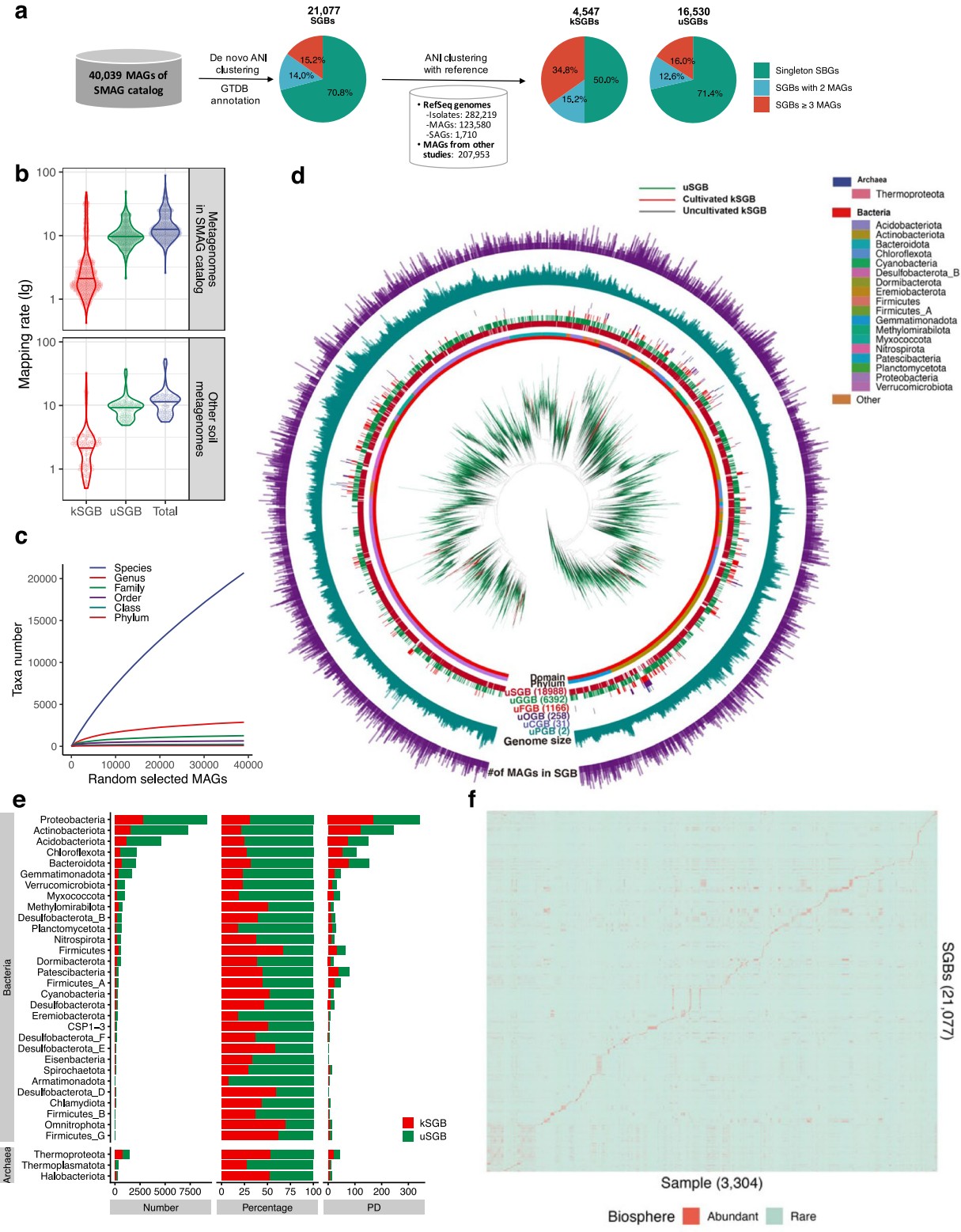

**Fig. 2 | The SMAG substantially expands the diversity of soil microbes. a** 16,530 genomes (41%) from SMAG (40,039 MAGs) were assigned to the uSGBs. **b** uSGBs improve mappability for soil metagenomes. **c** The rarefaction curve is obviously unsaturated at specie rank in the SMAG dataset. **d** A phylogenetic tree was built for 21,077 SGBs based on the concatenated 400 conserved universal PhyloPhlAn markers genes. **e** The comparison of the number of genomes across phyla between kSGBs and uSGBs. **f** The biosphere distribution of SGBs across metagenomic samples.

but with high proportion of core genes suggests its closed pan-genome and implies its critical role in fundamental functions in soils. Conversely, Nitrospirota was observed frequently in wastewater habitats[42], and had the lowest proportion of core genes, suggesting

its large pangenome openness which enables high environmental adaptability[43].

To investigate the functional divergence between core and accessory genes, we compared the proportion of genes assigned with

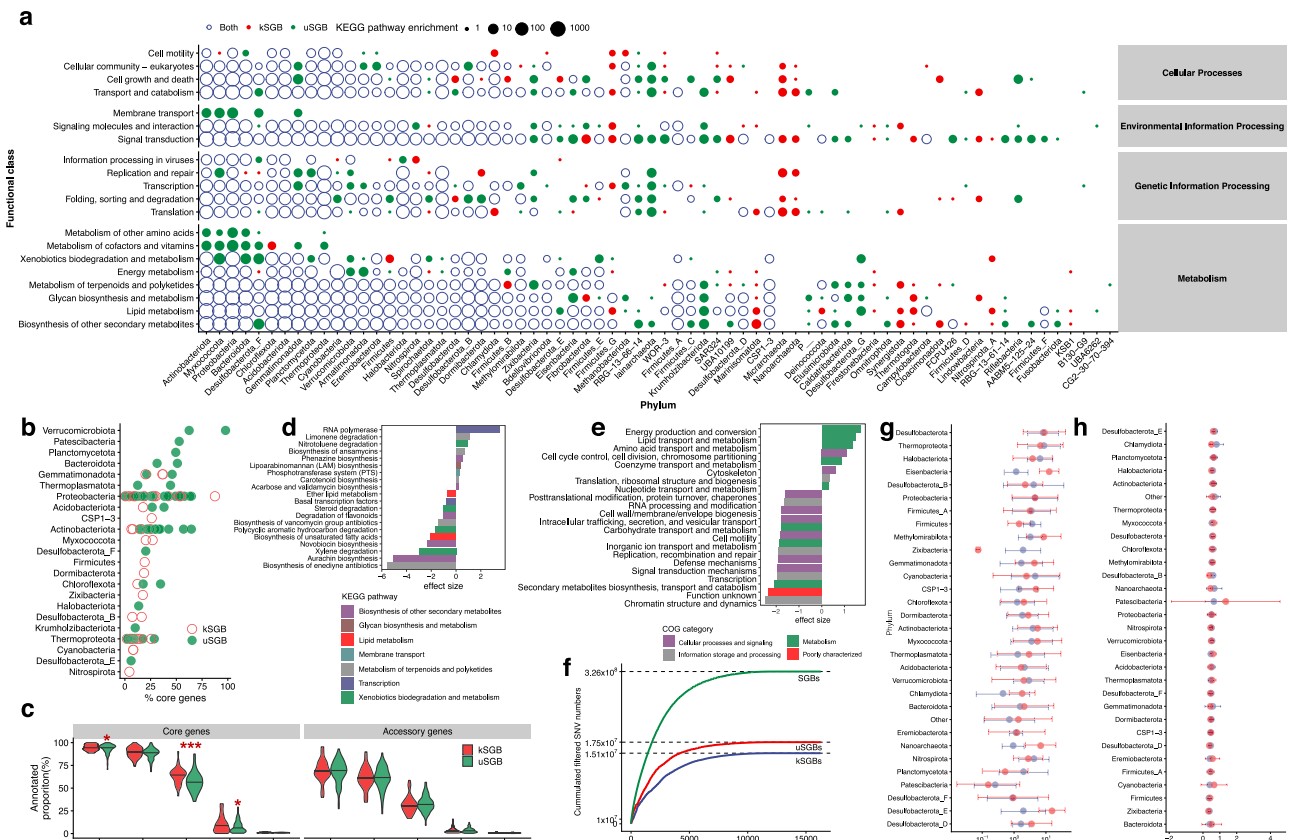

**Fig. 3 | Functional landscape and intraspecies genomic variation analyses within the soil microbiome. a** Functional category enrichment differential distribution between the kSGBs and uSGBs of the 5184 high-quality MAGs. uSGBs substantially expanded functional landscape of most of phyla in SMAG catalogue. **b** The abundance of core genes for kSGBs and uSGBs across phyla. **c** Proportion of core and accessory genes (*n* = 2,200 species) classified with various annotation schemes. A two-tailed Wilcoxon rank-sum test was performed to compare the classification between the core and accessory genes (*P < 0.05), eggNOG

(*P = 0.054), KEGG (***P = 0.0005), GO (*P = 0.041). **d** Comparison of the KEGG pathways between the core and accessory genes. **e** Comparison of the COG categories between the core and accessory genes. **f** Total number of SNVs detected as a function of the number of species, and uSGBs detected more SNVs than kSGBs. **g** The density of SNVs for kSGBs and uSGBs across dominant phyla (*n* = 2448 species). **h** The pN/pS ratios for kSGBs and uSGBs across dominant phyla (*n* = 2448 species). Data of (**g**) and (**h**) are presented as mean values +/− Standard Deviation (SD).

eggNOG in core and accessory genes. The core genes were better annotated than accessory genes based on all five databases (Wilcox test, *P* < 0.001), and the proportions of core genes of uSGBs annotated with eggNOG (Wilcox test, *P* = 0.054), KEGG (Wilcox test, *P* = 0.0005), and GO (Wilcox test, *P* = 0.041) were significantly lower than those of kSGBs (Fig. 3c). Thereafter, we investigated the functional enrichment by the core and accessory genes based on the eggNOG functional annotations. Significance was calculated with a two-tailed Wilcoxon rank-sum test and further adjusted for multiple comparisons using the Benjamini–Hochberg correction. A positive effect size (Cohen's d) indicates that the core gene is dominantly represented. The core genes were significantly assigned (*p* adjust < 0.001) to genetic information processing and key metabolic functions like Carotenoid biosynthesis and Phosphotransferase system (PTS). While great number of accessory genes are overrepresented in various secondary metabolites, such as Biosynthesis of enediyne antibiotics, Aurachin biosynthesis and Novobiocin biosynthesis with large effsize (d estimate >2), indicating the important role of accessory genes in defense activities (Fig. 3d). A similar tendency was found in the COG analysis. Core genes were dominantly represented in the basic cellular processes like Amino acid transport and metabolism. In contrast, more accessory genes are related to environmental adaptation and inter-strain differences. The accessory genes show dominantly represented in secondary metabolites biosynthesis, Transport and catabolism, Defense mechanisms. Moreover, a much greater proportion of function unknown COGs are

poorly characterized without a known function (Fig. 3e). These results provide a functional landscape difference between core and accessory genes identified from the SMAG catalogue by pangenome analysis.

To profile the intra-species variation of the SMAG catalogue, we investigated intraspecies single-nucleotide variants (SNVs) within SGBs with ≥3 MAGs. We detected 582,519,530 SNVs from 2448 SGBs with at least three conspecific MAGs (Fig. 3e, Supplementary Data 3). Of these SNVs, 326,163,258 (56%) filtered (exclude synonymous mutations) SNVs (were detected and 174,868,789 (53.6%) were found exclusively in uSGBs, and 151,294,469 (46.4%) were exclusively detected in kSGBs (Fig. 3e), indicating a large number of previously undiscovered SNVs in the SMAG catalogue. We also assigned the detected SNVs to the kSGBs and uSGBs across different phyla. Notably, we observed a divergence in the density of SNVs between kSGBs and uSGBs across most dominant phyla (Fig. 3g, Supplementary Fig. 3f). In addition, a majority of the phyla exhibited relatively low pN/pS ratios (pN/pS < 1) (Fig. 3h and Supplementary Data 3). This suggests that the evolution of soil microbial organisms might be more influenced by long-term purifying selection and drift, rather than by rapid adaptations to specific environments[44]. While species from Patescibacteria possess the smallest genome sizes, displayed the lowest SNV density coupled with the highest pN/pS ratios, possibly owing to their reduced redundant and non-essential functions that enable them to maintain community stability[34]. These findings suggest that the SMAG catalogue encompasses a significant amount of intraspecific SNVs. The observed

variations in SNV density and pN/pS ratios across different phyla underscore the diverse niche widths of these species and their varying capacities to acquire and allocate soil resources[45].

## Broad secondary metabolite biosynthetic potential

Microbial genomes encode biosynthetic gene clusters (BGCs) that produce natural secondary metabolites, offering vast potential for discovering ecologically and biotechnologically relevant enzymes and biochemical compounds. In addition to exploring BGCs from cultivated microorganisms[46], many studies have employed metagenomic data mining to survey BGCs for drug discovery[47] and microbiome ecology studies[9]. Given the tremendous microbial diversity in soil ecosystems, the SMAG catalogue offers an important resource for mining BGCs for natural product development and drug synthesis. We identified 70,081 putative BGCs, of which 69,990 were annotated with one or more BGC types. The BGCs identified from the 21,077 representative MAGs of the SMAG catalogue are 36 times the number of the manually curated Minimum Information about a Biosynthetic Gene (MIBiG) dataset[46]. After filtering contigs ≥ 5 kb, 43,169 BGCs were categorized into eight groups (Supplementary Data 4), most of which were identified from uSGBs (Fig. 4a). The number of non-ribosomal peptide synthetase (NRPS), the necessary multienzyme machinery for assembling numerous peptides for antibacterial (such as penicillin)[48], was the highest with a total of 10,277 (23.8%) BGCs encoded by 49 phyla. We also identified 9632 (22.3%) BGCs synthesizing ribosomally synthesized and post-translationally modified peptide (RiPPs) from 69 phyla, 7671 (17.8%) terpene gene clusters from 45 phyla, 1790 (4.1%) polyketide synthase (PKSI) clusters from 28 phyla, and 1664 (3.9%) PKS–NRPS hybrid gene clusters from 23 phyla (Fig. 4c).

We then assessed the biosynthetic potential of the dominant phyla (Fig. 4b). Consistent with the GEM catalogue[29] and glacier catalogue[9], Proteobacteria process the greatest biosynthetic potential, with 1439 NRPS, 2153 RiPPs, 2052 terpene, and 3216 other BGCs encoded by 6774 Proteobacterial MAGs, followed by Actinobacteriota with 5376 MAGs encoding 9575 BGCs. Furthermore, we identified a total of 9119 BGCs encoded by 4781 Acidobacteriota MAGs, with one MAG from unannotated genus of family UBA5704 encoding 111 NRPS or PKS modules with clear colinear module chains (Fig. 4d). In addition, we identified high biosynthetic potential for Gemmatimonadota (2633 regions across 1300 MAGs), indicating we may underestimate the biosynthetic potential of these linkages. Although most identified BGCs were fragmented (Supplementary Fig. 4a), we identified 742 regions with a length >50 kb and 4772 regions >30 kb. Five NRPS clusters with a length >100 kb (Supplementary Fig. 4b–f) were all identified from uSGBs. The largest BGC in the SMAG catalogue (270,820 bp) was identified from genus UBA5704 of Acidobacteriota (Fig. 4e), while the largest BGC (275,339 bp) from GEM was identified from the same genus but with the core biosynthetic sequence identity range of 0–52.48% (Supplementary Data 4). We found that both of the two BGCs were mainly involved in amino acid metabolism, but SMAG_BGC exhibited an additional involvement in carbohydrate metabolism and environmental information processing based on the KO assignment results (Fig. 4f). Taken together, these results suggest that the SMAG catalogue can serve as a valuable resource for the discovery of new drugs and therapeutics.

## CRISPR and Cas protein genetic resources

Microbes rely on diverse defense mechanisms that allow them to withstand viral predation and exposure to foreign DNA. Many Bacteria and Archaea possess clustered regularly interspaced short palindromic repeats (CRISPR) together with CRISPR-associated genes (Cas), called CRISPR–Cas systems, to prevent viral infection[49]. Spacers are the regions of the leader end of the CRISPR array with a length of 24–48 nucleotides[50] to be transcribed and processed into CRISPR RNAs (crRNAs) for the microbes' "immune" system[51]. The SMAG catalogue

affords a significant opportunity to explore the diversity of Cas proteins resources for transforming and synthesis efficient gene editing tools.

To profile the diversity of genetic resources associated with CRISPR-Cas systems in the SMAG catalogue, we characterized spacers and Cas genes by predicting open reading frames (ORFs) and aligning to Cas proteins in National Center for Biotechnology Information (NCBI). In total, we identified 1142 spacers from 662 MAGs (Supplementary Fig. 5a, Supplementary Data 5), on average $0.40 \pm 0.35$ (mean ± SD) spacer sequences per Mb of genomic length. Given the number of spacers in each MAGs displayed on a scale-free distribution (Fig. 5a), the majority (454) of MAGs possessed only one spacer sequence and a few MAGs possessed more than 10 spacer sequences, indicating their potential ability to defend against viral infection. The number of spacers did not increase with genome size (Fig. 5b). MAGs with a genome size of ~5.5 Mb, either from kSGBs or uSGBs, possessed the highest number of spacers, but there was no difference in the source between kSGBs or uSGBs (Supplementary Fig. 5b). Spacer loads differed significantly across phyla (Fig. 5c), with the highest density of spacer loads for Cyanobacteria and Proteobacteria, on account of the largest number of genomes, and with the lowest density of spacer loads for Firmicute_E and Myxococcota. The largest numbers of spacers in MAGs were found in a cyanobacterial MAG reconstructed from grassland that possessed 20 spacer sequences. This could be explained by the fact that Cyanobacteria is the only bacteria with a rich number of transposable elements and transposase genes involved in the complex differentiation process[52].

We further quantified 8545 Cas-associated genes from 563 MAGs, with an average of $281.2 \pm 219.6$ Cas-associated genes per MAG. (Supplementary Fig. 5c, Supplementary Data 5). Cas1 and Cas 2, are highly conserved, generally as a universal marker for CRISPR-Cas systems[53], which was the most widely-known protein identified from the SMAG catalogue. Approximately 200 Cas 2 were identified for small putative nucleases (80–120 aa) and considered a second marker for CRISPR-Cas systems[54]. Notably, we identified 42 Cas 9, which were potentially engineered for powerful genome editing tools[55]. 245 MAGs (43.5%) possessed less than 10 Cas-associated genes (Fig. 5d) and only 1611 Cas-associated genes (18.8%) were identified with certain Cas-associated genes (Fig. 5g, Supplementary Fig. 5d). The collection of Cas protein family profiles is a resource for the identification of CRISPR–Cas systems[3], which also illustrates the necessity and importance of mining the soil microbiome.

MAG with the largest number of spacers (20) also possessed a large number of Cas-associated genes (238) (Supplementary Fig. 5a, c). Consistent with spacers, MAGs with a genome size of ~5.5 Mb, either from kSGBs or uSGBs, had the highest numbers of Cas-associated genes (Fig. 5e). The density of Cas-associated genes in genomes varied with various phyla (Fig. 5f), with the highest density in Patescibacteria and Firmicute_C, and the lowest density of in Myxococcota and Desulfobacterota_F. uSGBs expanded the profiles of Cas protein resources in many phyla, such as Verrucomicrobiota, Armatimonadota, Gemmatimonadota, Fusobacteriota and Desulfobacterota_B (Fig. 5h), indicating that uSGBs offered an important information about Cas proteins from the soil microbiome. This also demonstrates the utility of metagenomic mining for gene editing tools development.

## Connecting MAGs to virus-host associations

Previously uncharacterized MAGs help to improve predictions of virus-host association prediction, which are crucial for understanding the roles and impacts of viruses in natural ecosystems. In this study, 21,510 virus-host associations were identified by predicting prophages. (Supplementary Data 6). Those prophages can be clustered into 257 clusters at the family level. The predicted virus-host associations were mainly contributed by Actinobacteria (8116), followed by Proteobacteria (3468), Acidobacteria (3162), and Thermoproteota (2310)

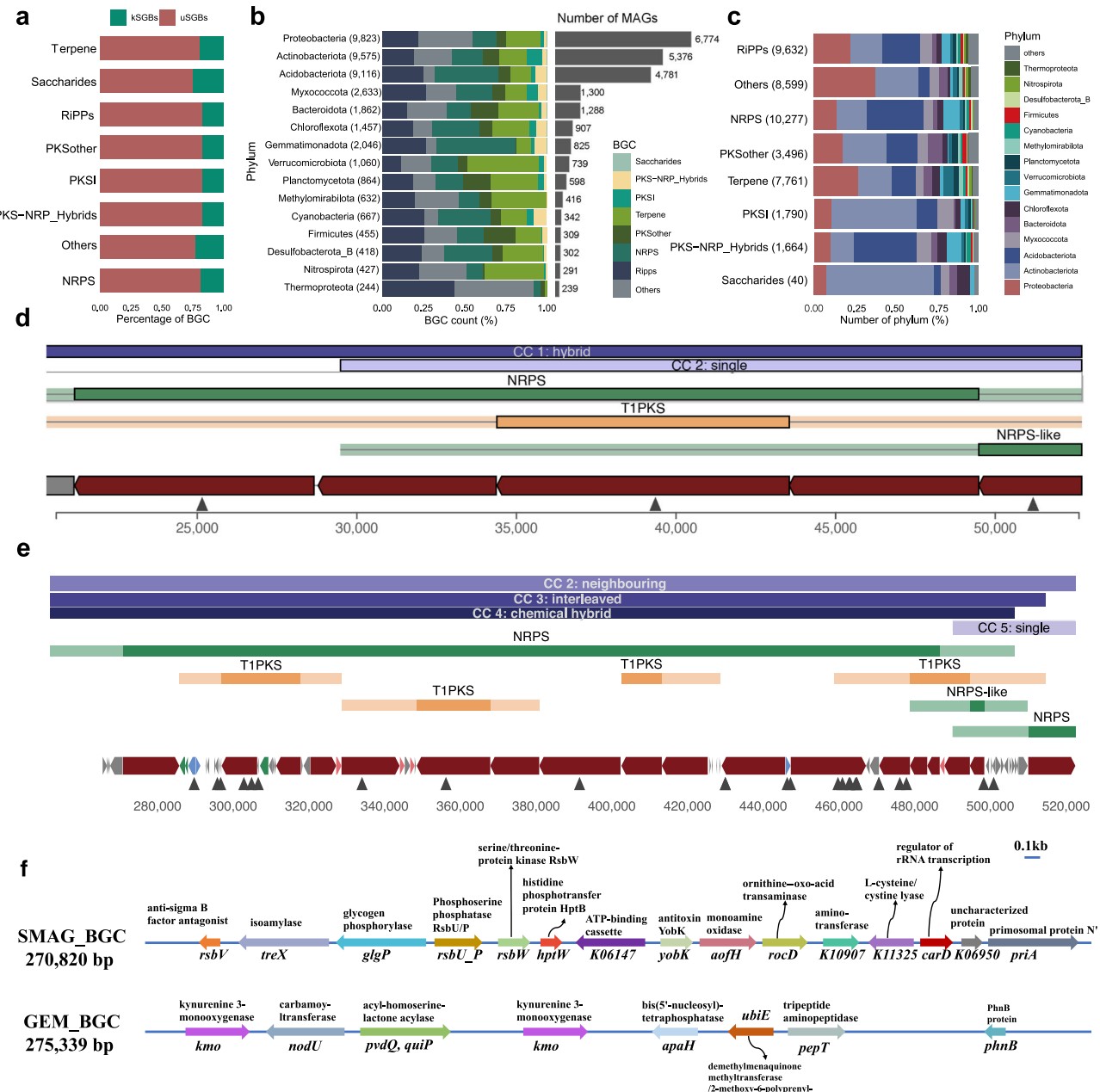

**Fig. 4 | Biosynthetic gene clusters recovered from the SMAG catalogue. a** BGCs of the SMAG between kSGBs and uSGBs. All the BGCs were separated into eight BiG-SCAPE classes. Non-ribosomal peptide synthetase (NRPS), Ribosomally synthesized and post-translationally modified peptide (RiPPs), polyketide synthase (PKS I), Terpene, PKS–NRPS hybrid, PKS other, Saccharides, Others. **b** The relative frequency of BGC types across dominant phyla BGC genes are predominantly identified in Proteobacteria, Actinobacteriota, Acidobacteriota and Bacteroidota. They are highly variable across phyla. **c** Number and BGC types identified from the SMAG. **d** Encoding the most remarkable number of BGC clusters including 111 NRPS or PKS modules and with clear colinear module chains. **e** The single largest BGC region found in a soil-derived bacterium from the Acidobacteria phylum and UBA5704 family. **f** Distribution and KO assignment of the two largest BGCs from SMAG and GEM.

(Fig. 6a). The proportion of the uSGBs contributed to virus–host associations was 84.6% (Fig. 6b), suggesting that uSGBs from the SMAG catalogue considerably expand our understanding of virus-host associations.

To explore the host phylogenetic ranges of viruses, we analyzed the host taxa of 76 generalist viruses with >25 predicted hosts (The term "generalist" viruses refer to the potential host range of a virus). Many studies indicate that viruses can alter the host metabolic process and participate in element cycling in the soil through a variety of auxiliary metabolic genes[56]. Most of those generalist viruses are mainly predicted from uSGBs, indicating uSGBs from the SMAG catalogue reveal a great deal of previously unexplored virus–host associations involved in the geochemical cycles of the global soil elements. However, the proportion of kSGBs hosts was >88% for acidobacteriotal virus GSV_39462, proteobacterial virus GSV_66726, and patescibacterial virus GSV_270. Almost all of those generalist viruses predicted potential hosts from the same phyla except GSV_42450 (Fig. 6c), which was predicted from MAGs that ranged from nine phyla, mainly from Acidobacteria (33 SGBs), Gemmatimonadota (seven SGBs), and Actinobacteriota (six SGBs).

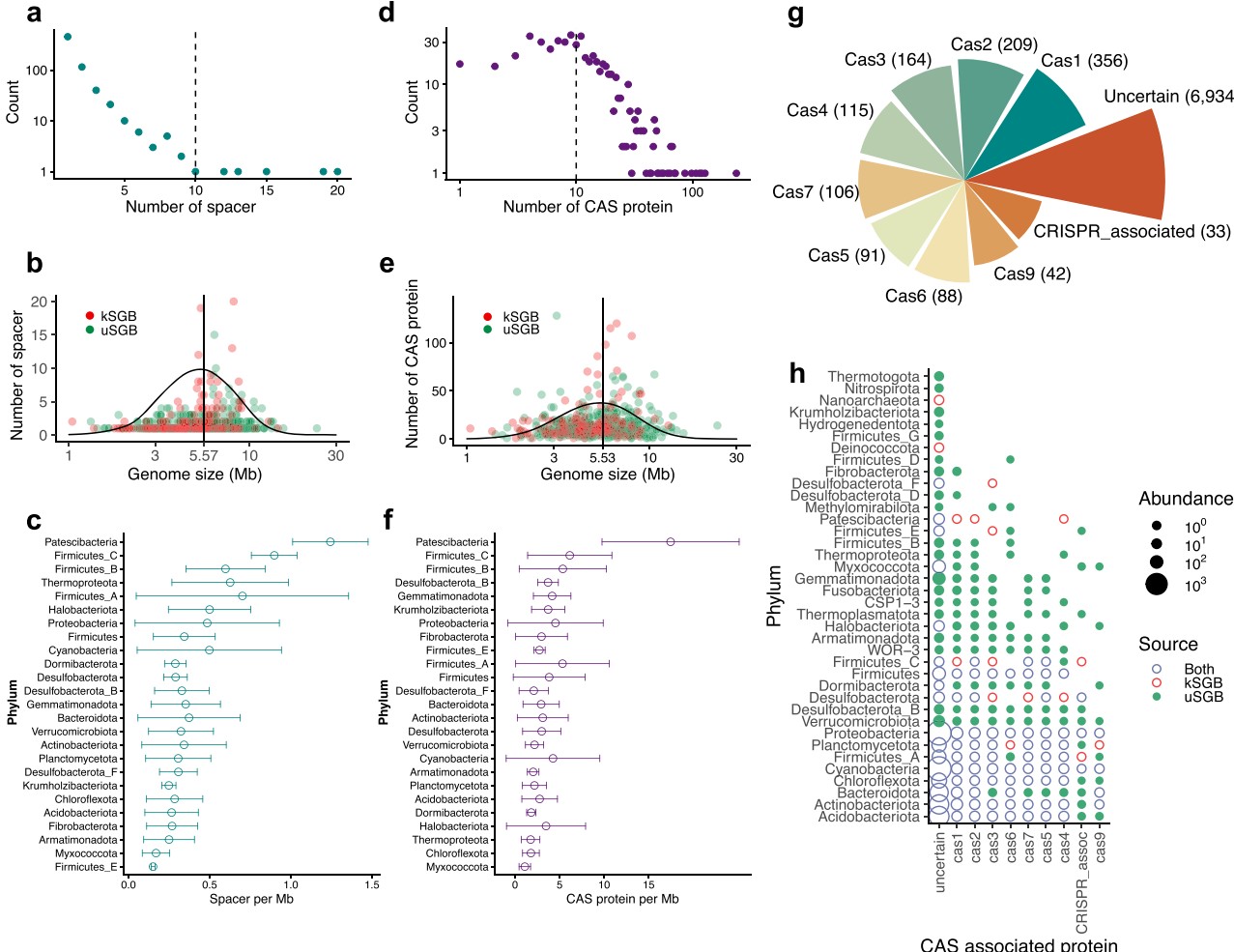

**Fig. 5 | The profile of spacers and Cas-associated proteins in the genomes of SMAG catalogue. a** Most MAGs possessed only one spacer sequence. **b**, **e** The number of spacer sequences and Cas-associated genes did not increase with genome sizes, either for kSGBs or uSGBs. **c**, **f** Spacer sequences (*n* = 662 MAGs) and Cas-associated gene (*n* = 563 MAGs) loads differed significantly across phyla, data of (**c**) and (**f**) are presented as mean values +/− SD. **d** The count of Cas-associated genes processed among MAGs. **g** The top 10 number of different Cas proteins processed from the SMAG catalogue. Most (6934) of predicted Cas genes were uncertain. **h** The profile of Cas-associated genes processed for kSGBs and uSGBs across phyla. uSGBs expanded the profiles of Cas-associated genes.

## Discussion

Here we established the SMAG catalogue by reconstructing 40,039 bacterial and archaeal genomes, representing 21,077 species-level genome bins, from large-scale metagenomic assembly. As a result of our work, the majority (16,530 uSGBs) of reconstructed genomes are currently unidentified from species to class level, and uSGBs made a great contribution to increasing the mapping rate of soil metagenomes.

We found the uSGBs immensely expanded the functional landscape of soil microbiota. The pangenome (107) and SNV (582,519,530) analyses show a large number of unknown core genes that need further investigation. Based on the proportion of core genes, we identified the divergence of pangenome openness across different phyla, suggesting their divergent roles in ecological functions in soils[43]. These results indicated that pangenome evolution analysis within defined phylogenetic groups should consider the environmental effect[41]. A large number of SNVs were detected from uSGBs in the SMAG catalogue, revealing a lot of previously unknown intraspecies variations. And the divergent pN/pS ratios indicated the soil microbiome experienced a strong purifying selection, which may highlight the environmental adaptability of species within the community, emphasizing a balance where deleterious genetic variations are minimized[45].

The SMAG catalogue showed a rich discovery potential for BGC diversity, which is a vital resource for the synthesis of natural products[57]. We found most BGCs identified were from genomes of uSGBs and the biosynthetic potential of microorganisms is divergent across various BGC types, indicating the great potential for mining previously unexplored BGCs from the uncultivated and unknown microorganisms from soils. However, most BGCs identified from the SMAG catalogue were fragmented, indicating that short-read sequencing restrained the recovery of full-length BGC sequences from uncultivated bacteria[15], and tools based on Hidden Markov Model (HMM)-based algorithms limited the accuracy and generalizability of BGC identification[58]. Long-read sequencing and deep learning adopted for metagenomic assembly may enable more complete genomes[59], high-resolution analysis of resistance determinants and mobile elements[60]. Thus, future research can combine the long-read sequencing to construct more complete BGCs[15] and machine learning can be introduced into the identification of the BGCs region[61] and the mining of microbial dark matter[60].

The SMAG catalogue also encompassed great potential in the development of anti-viral defense systems. We detected 8545 natural CRISPR-Cas genes, revealing the considerable potential of the SMAG catalogue for mining gene editing tools. We identified that uSGBs offered plenty of previously unexplored resources of Cas proteins

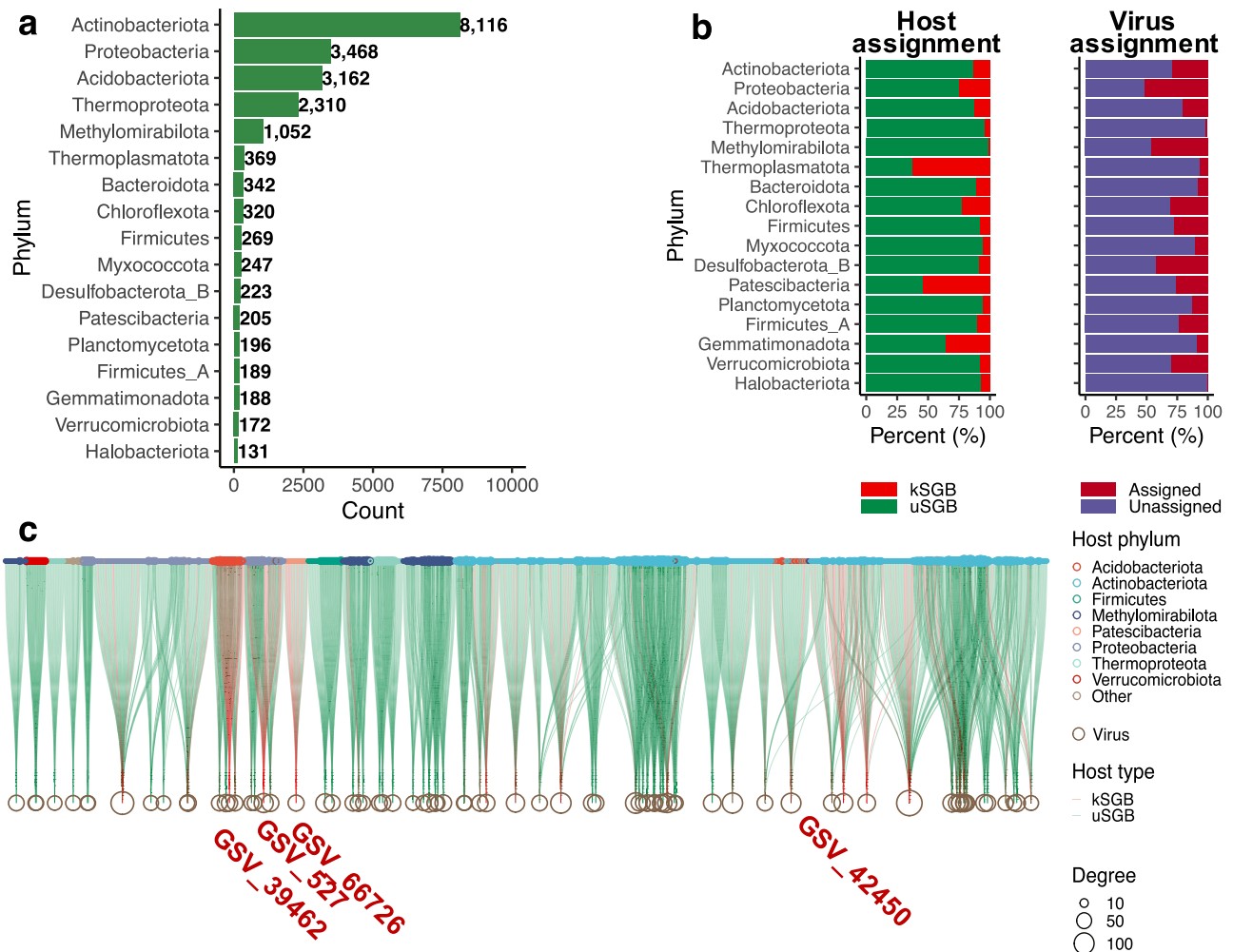

**Fig. 6 | The SMAG resolves virus-host connectivity. a** The virus-host association counts across phyla. **b** The virus-host associations for kSGBs and uSGBs predicted by prophages. **c** The host phylogenetic ranges of viruses. GSV_66726, GSV_527, and GSV_39462 were the previously unidentified virus from in-house data Global Soil Virome (GSV).

resource. Different phyla showed varied potential for developing "immune" systems based on the spacers and Cas gene numbers, which can guide researchers to mine targeted "immune" systems[53]. Furthermore, the identification of spacers helps to understand the process of insertion of spacer sequences into the host CRISPR locus to generate immunological memory[62]. Overall, the analysis makes the first large-scale and comprehensive portrayal of the Cas proteins resource in the soil microbiome, which is a momentous resource for exploring the molecular "immune" system of microbes.

The uSGBs contributed the most virus-host associations in the SMAG catalogue. Most virus-host associations were predicted by prophages, a prevalent infection pathway of viruses in soil microbiota[21]. We found divergent host phylogenetic ranges of viruses across different phyla. Interestingly, the finding of a generalist virus offers new insights into experimental work of phage cultivation[63]. Together, these results demonstrate previously unexplored putative virus–host connections, expanding our understanding of soil microbial dark matter.

In summary, we have established this soil MAGs catalogue, which sheds light on soil microbial dark matter and provides valuable insights into the diversity and function of soil microbiomes. Besides, given the large uncultured and unknown diversity remaining in soil microbiomes, highlighting major informational databases for provocative new biological insights and having a high-quality genome catalogue substantially enhances the resolution and accuracy of metagenome-based studies for the broad relative readership. Also, the MAGs in the SMAG catalogue are resources for building genome-scale metabolic models (GEMs), which would be a crucial resource for designing and engineering microbiomes. All in all, knowledge gained from this work is valuable as a genetic resource for future studies based on genome-centric mining, and to prioritize targets for further experimental validation.

## Methods

### Sampling, sequencing, and collection of soil metagenomes

We downloaded 2941 soil metagenomes from the NCBI Sequence Read Archive (SRA)[64] publicly available with file sizes exceeding 2GB from different countries which cover 9 soil ecosystems and about 363 in-house data were sampled, see details in Supplementary Data 1.

We collected ecosystem classifications manually, and for projects with insufficient information, we defined the ecosystem type by latitude and longitude using GlobeLand30 and Google Maps In-house samples from this study were sampled by our team across China (348) and Europe (15) in 2018–2020 using a standard sampling protocol[65]. Five-point sampling method (non-probability sampling) was performed in house samples. All soil samples were kept cool using dry ice until visible roots and stones were removed. And then all clean soils were stored at −80 °C until DNA extraction. In all cases, DNA extraction of 400 mg of soil in each sample was performed using MP FastDNA SPIN Kits 385 for soil (MP Biomedicals, Solon, OH, USA) according to

the manufacturer's instructions and DNA was purified and concentrated using Qubit fluorometric quantitation (Thermo Fisher Scientific, 388 Waltham, MA, USA). Purified DNA was stored at −20 °C for sequencing. Metagenomic sequencing from each soil sample was conducted by Illumina HiSeq 4000 or Illumina novaseq pe150 (Illumina, San Diego, CA, USA), generating 150 bp paired end reads. Sequence data have been deposited in the public NCBI under BioProject accession numbers PRJNA983538.

## Metagenome quality control, assembly
All the downloaded SRA files were split into paired-end raw reads using fastq-dump (v2.9.6) from sratoolkit (v2.9.6) with option '−split-3', and then all raw reads were separately quality-controlled using Trimmomatic[66] (v2.39) to trim adaptors and primers, and to filter short (<50 bp) and low-quality reads (<20 bases), followed by assembly with MEGAHIT[67] (v1.2.9) with a minimum contig length of 500 bp and with the options '--k-step 10 --k-min 27' to each sample separately.

## Metagenome binning and refinement
Soil MAGs were recovered for individual metagenomic assemblies using metaWRAP[68] on the basis of tetranucleotide frequencies (TNF) and coverage information, contigs shorter than 1000 bp were discarded. The resulting MAGs were refined using the module 'bin_refinement' from metaWRAP[68] (v1.2.1) to combine and improve the results generated by the three binners. During refinement, the completeness and contamination of all MAGs were estimated using CheckM[69] (v1.0.11) via the lineage-specific workflow with the options '-c 50 -x 10' to filter MAGs to be at least 50% complete, with <10% contamination. Ribosomal RNAs (rRNAs) were identified with nhmmer function (part of HMMER 3) from Barrnap (v.0.9) with the options '-reject 0.01 −e-value 1e-3' and '-kingdom bac/arc' for bacteria and archaea, respectively. Transfer RNAs (tRNAs) were annotated with tRNAscan-SE[70] (v.2.0.9) with options '-A' for archaeal species and '-B' for bacterial lineages. Based on these results, we classified the MAGs as the high quality based on the MIMAG standard[24] (>90% completeness, ≤5% contamination, ≥18/20 tRNA genes, and the presence of 5S, 16S, and 23S rRNA genes), with the remaining classified as medium quality.

## Dereplication and species-level genome bins clustering of SMAG
The 40,349 MAGs from the SMAG dataset were further quality-filtered with the function '--checkM_method (lineage_wf)' to avoid low-quality genomes, and then the 40,039 filtered MAGs were dereplicated and clustered into 21,077 SGBs based on 95% ANI with the following options: '-pa 0.9 -sa 0.95 -nc 0.10 -cm larger' using dRep[71] (v2.2.4). To reduce the computational burden of clustering all genomes, we used the multi-round clustering method just by set the parameter '--multiround_primary_clustering' from dRep which is helpful when clustering 5000+ genomes and will be done with single linkage clustering aiming to reduce the final computational load which was previously used to cluster >200,000 human gut MAGs[13].

## Phylogenetic and taxonomic annotation of SMAG
A total of representative 21,077 SGBs were classified with GTDB-TK[72] (v.1.6.0) using 'classify_wf' function and default parameters according to the Genome Taxonomy Database (GTDB) (release 202)[37]. In short, the GTDB-Tk classifies each genome based on ANI to a curated collection of reference genomes, placement in the bacterial or archaeal reference genome tree, and relative evolutionary distance (RED). The phylogenetic analyses of 21,077 SGBs were performed with PhyloPhlAn[73] (v3.0.60). The phylogeny in Fig. 2 was built using the 400 universal PhyloPhlAn markers with the following options: '--diversity high --accurate --min_num_markers 100'. For the internal steps the following tools with their set of parameters were used: Diamond[74] (v0.9.14.115) with parameters: 'blastp --quiet --threads 1 --outfmt 6 --more-sensitive −id 50 --max-hsps 35 -k 0'; mafft[75] (v7.310) with the

'--anysymbol' option; trimal[76] (v1.4rev15) with the '-gappyout' option; FastTree[77] (v2.1.10) with '-mlacc 2 -slownni -spr 4 -fastest -mlnni 4 -no2nd -gtr -nt' options; RAxML[78] (v8.1.12) with parameters: '-m PROTCATLG -p 1989 <phylogenetic tree computed by FastTree >.' and the best tree refined by RAxML is visualized ggtree[79] (v3.2.1).

To estimate the relative abundance of each MAG from separate soil samples, clean reads of each sample were aligned to the SMAG catalogue after de-replicating all MAGs at 95% identity with dRep[71] (v2.2.4) to avoid arbitrary mapping between representatives of highly similar genomes using BWA[80] (v0.7.17). The outputs were converted to BAM format by Samtools[81] (v1.10). Then the BAM was filtered with coverM v0.2.0 (https://github.com/wwood/CoverM) with the options "--min-read-percent-identity 0.95 --min-read-aligned-percent 0.90", the coverage of each contig was calculated with coverM using 'trimmed_mean' mode, so calculating the coverage as the mean of the number of reads aligned to each position, with the 10% smallest fraction of positions and 90% maximum fraction for trimmed_mean calculations. The coverage of each MAG was calculated as the average of contig coverages, weighting each contig by its length in base pairs. The relative abundance of each lineage in each sample was calculated as its coverage divided by the total coverage of all genomes in the dereplicated set. And samples with relative abundance of mag <0.01% were considered as rare biosphere, otherwise they were considered as abundant biosphere[36].

## Comparing MAGs to >500,000 genomes in public databases
We compared representative genomes from the 21,077 SGBs to a large number of publicly available reference genomes. Approximately 500,000 reference genomes were obtained from a variety of sources, including NCBI RefSeq (n = 282,219), GenBank (123,580 MAGs and 1710 SAGs) of November 2021 and multiple system-associated MAGs from several recent studies (207,593)[29,30,82]. We first used Mash[83] (v2.3) with the function of 'dist' to find the most similar reference genome to each of the 21,077 SGBs, and then we used the MUMmer[84] (v4.0.0) with the function 'dnadiff' and default parameters to estimate ANI between genome pairs. Based on the analysis results, a species was considered to have been cultured if it matched an isolate RefSeq genome with at least 95% ANI over at least 30% of the genome length, and we considered a species as an unknown genome if it represented only by SMAG.

## Functional analysis of SMAG
And putative protein-coding sequences (CDSs) of SMAG were predicted using Prodigal[85] (v2.6.3) with the '-p single' parameter. The predicted CDSs were then clustered by MMseqs2[86] with the options'--min-seq-id 0.95 -c 0.9 --cluster-mode 2 --cov-mode 1', and then the representative CDSs were annotated with eggNOG-mapper[87] (v2.1.6) with database (v5.0)[38], and KEGG (Kyoto Encyclopedia of Genes and Genome) and Clusters of Orthologous Groups of proteins (COGs) functional annotations were derived from the eggNOG-mapper results.

## The secondary-metabolite biosynthetic potential of SMAG
Secondary-metabolite BGCs of SMAG were identified using antiSMASH[58] (v6.1) with default settings and the corresponding database (v5.0)[88], Then BGCs were subsequently filtered, retaining only the ones encoded on scaffolds ≥5 kb to reduce the risk of fragmentation, as done previously[29,89], which resulted in a total of 43,169 BGCs (Supplementary Data 5). And these BGCs were categorized into eight groups: 'PKSI', 'PKS-NPR_Hybrids', 'PKSothers', 'NRPS', 'RiPPs', 'Terpene', 'Saccharides' and 'Others', based on the categories suggested by the BiG-SCAPE[90]. We selected sequences encoding core biosynthetic genes from the two BGCs to do sequence identity comparison by Clustal (v2.1)[91], and the KO of the largest BGCs from SMAG and GEM were assigned using BlastKOALA[92]v.2.21.

## SNV, Pangenome analysis of SMAG

A total of 2448 species with at least three conspecific genomes (completeness >= 50%, contamination <= 5%) were used to generate a catalogue of SNVs (Supplementary Data 3). We mapped all conspecific genomes to the representative genome for each species using the 'nucmer' program from MUMmer[84] (v4.0.0) and filtered alignments using the 'delta-filter' program with options '-q -r' to exclude chance- and repeat-induced alignments. Thereafter, we identified SNVs using the 'show-snps' program. Single-base insertions and deletions were not counted as SNVs. Each SNV locus was included in the catalogue only when the alternate allele was detected in at least two conspecific genomes. To filter the synonymous SNVs, we calculated the synonymous ratio with the house script snv-filter.py, and we estimated the ratio of non-synonymous to synonymous polymorphism rates[44] (pN/pS) to evaluate the genetic diversity. Pan-genome analyses were carried out by selecting 2200 SGBs with >10 high-quality MAGs (completeness >= 80%, contamination <= 5%) using Roary[93] (v3.12.0), with the options of a minimum amino acid identity at 90% ('-i 90') and a core gene defined at 90% presence ('-cd 90').

## CRISPR and Cas protein

CRISPR arrays were identified on contigs longer than 3 kb in MAGs using a combination of PLIER-CR[94] (v0.4.2). And the MAGs containing fewer than four CRISPR-associated proteins were removed. Proteins were predicted with Prodigal (v2.6.3) and de-duplicated to construct a database. Proteins with lengths between 200 and 1000 aa were obtained. The NR database was used to remove proteins of known function and Cas proteins in NCBI were used for further characterization of the candidate Cas proteins.

## Connecting MAGs to viruses identified from VirSorter2

To maximize the number of prophages identified in MAGs, we used VirSorter2[95] (v2.0.alpha) to perform de novo prediction. Only those classified into prophage by CheckV[96] (Version 1.0) were retained. To exclude possible decayed prophages, that is, integrated virus genomes which are now inactive and progressively removed from the host genome, all predictions for which 30% or more of the genes displaying the best hit to Pfam (35.0)[97] were excluded (thresholds: hmmsearch score ≥ 50 and E ≤ 0.001).

## Statistics and reproducibility

No data were excluded from the analyses. The investigators were not blinded to allocation during experiments and outcome assessment.

## Reporting summary

Further information on research design is available in the Nature Portfolio Reporting Summary linked to this article.

## Code availability

The workflow used to generate the genome, taxonomic analysis, and functional annotation, alongside the BGCs, pan-genome, SNV annotations, and virus predictions and scripts used to generate the figures are described at GitHub repository through https://github.com/Caiyulu-818/SMAG/releases/tag/v1.0 (ref. 98). All statistical analyses for generating figures were performed using the R environment v4.1.2[99].

## Data availability

The raw sequence data of the in-house samples reported in this paper and the 16,530 uSGBs of the SMAG catalogue have been deposited to NCBI SRA and GenBank under the bioproject accession number: PRJNA983538. For the bulk download, all the MAGs, SNV catalogues and viruses predicted the SMAG has been deposited in Zenodo repository through https://doi.org/10.5281/zenodo.7341719 (ref. 100) and also be available in the freely accessible interface-web of the SMAG catalogue (https://smag.microbmalab.cn). The source data underlying Figs. 1–6 and Supplementary Figs. 1-6 are provided as Source Data files and have been deposited in the Figshare database (https://doi.org/10.6084/m9.figshare.23298791). The databases used in this study include GEM catalog (https://genome.jgi.doe.gov/portal/GEMs/GEMs.home.html), the UHGG (https://ftp.ebi.ac.uk/pub/databases/metagenomics/mgnify_genomes/), and GTDB database Release 202 (https://data.ace.uq.edu.au/public/gtdb/data/releases/release202/).

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

## Acknowledgements

We thank C. Kelly, C. Averill, D. Buckley, D. Goodheart, D. Duncan, D. Myrold, E. Eloe-Fadrosh, E. Brodie, E. Högfors-Rönnholm, H. Cadillo-Quiroz, J. Tiedje, J. Jansson, J. Norton, J. Blanchard, J. Schweitzer, J. Banfield, J. Gladden, J. Raff, K. Peay, K. Gravuer, K. M. DeAngelis, L. Meredith, M. Kalyuzhnaya, M. Waldrop, N. Fierer, P. Dijkstra, P. Baldrian, S. Theroux, S. Tringe, T. Woyke, T. Whitman, W. Mohn & San Diego State University for their permission to use their metagenome data. We also thank Jianyu Jiao from Sun Yat-Sen University for the useful discussion. Thanks for the support from Amazon Web Services for providing computing resources. This work was supported by the National Natural Science Foundation of China (grants 42090060, 42277283 to B.M. and 41991334 to J.X.), the Key R&D Program of Zhejiang Province (2023C02004 to B.M. and J.X., 2023C02015 to B.M.), and the Fundamental Research Funds for the Central Universities (226-2022-00139 to B.M.).

## Author contributions

B.M., J.X., J.Z., and Y.Z. conceived and co-supervised the study. C.L., Y.W., J.Y., K.Z., R.X., and H.R. designed the methods of the research. B.M., C.L., Y.W., J.Y., X.L., R.X., R.P., J.Z., Y.Z., and J.X. performed bioinformatic and statistical analyses. B.M. and C.L. drafted the manuscript; B.M., C.L., X.L., R.P, J.Z., Y.Z., and J.X. reviewed and edited the manuscript. B.M. and C.L. performed funding acquisition.

## Competing interests

The authors declare no competing interests.
