## [Peer Review File · Nature Communications]

REVIEWER COMMENTS

Reviewer #1 (Remarks to the Author):

The authors put together a global database of soil metagenomes and assemble thousands of genomes to characterize the functional profiles of soil prokaryotes. The study represents a new catalogue of soil MAGs and their functional characteristics.

I believe this study has a huge amount of work and deserve publication. However, I also found some significant limitations and details that need to be addressed before publication:

1. There is a growing number of soil MAGs catalogues being published in the literature. The study claims that this is “the first excavation of soil microbial dark matter”. However, a recent study published in Nature also included a global characterization of soil MAGs (and those of other habitats) (<https://www.nature.com/articles/s41586-021-04233-4>). What is your study adding to these previous studies?

2. In the abstract, it is highlighted that 40,039 MAGs were assembled. Yet, only 3641 supported enough quality (>90% completeness and <5% contamination) to be explored. I think that all functional analyses need to focus on these 3641 MAGs, as the other MAGs are not complete enough to provide reliable information. It is not clear to me if functional analyses were done with all 40,039 MAGs, or only with those showing enough quality. Also, how many of these MAGs are non-redundant?

3. More information on the precedence of these soil metagenomes needs to be provided in a supplementary table. Are these samples from topsoil? Lines 628-633 are difficult to follow. What are in-house samples? Where are those samples coming from? Ecosystems from China?

4. The title of the paper and the abstract could highlight that this study includes a new catalogue of soil MAGs and their functional profiles.

Reviewer #2 (Remarks to the Author):

Comments to the Author

In the manuscript “Soil microbial dark matter explored from genome-resolved metagenomics”, Ma et al. have reconstructed a huge number of metagenome-assembled genomes from about 3000 soil metagenomes with a large proportion of unknown species-level genome bins. This study has expanded archaeal and bacterial genome diversity across the tree of life and enlarged the genomic database, which is very useful for the further study of genetic resources. I appreciate that authors have performed intensive sampling and bioinformatic analyses. I have some comments and suggestions for authors which help to revise the manuscript.

L100: should list ref 25 here as well.

L151: “kSGBs” should be described when it first appears (L135?).

L184: the authors may consider proposing new taxonomic levels based on GTDB and SeqCode1 rather than 16S rRNA. Also, “16s” should be “16S”.

Line 206 to 209: Due to the large number of uSGBs, the higher gene number in uSGBs seems predictable. The proportion of each COG functional category may be suited to studying the function difference between uSGBs and kSGBs.

L214: Could the authors clarify the clustering method? Why chose 90% amino acid identity as the threshold?

Extended Data Fig. 5d: “Number of cas protein” should be “Proportion of cas protein”.

L266-267: Have the authors considered using adding dN/dS ratio rather than SNV alone as proxies for connecting genetic diversity to ecosystem functions? Because synonymous mutations are neutral and do not contribute to changes in protein functions. Thus, this part of SNV (synonymous) may not lead to niche breath changes and adaptability to different environments for certain microorganisms.

L319-321: Are there any results support that the identified BGCs in this study were novel compared to existing studies? Since the authors suggest that BGCs found in this study may lead to the development of new drugs and therapeutics.

L364: "cas" should be capitalized as "Cas" throughout the text.

L369: What's before "of which"?

L372-374: similar to BGCs, what specific results indicate that the finding in this study may facilitate the development of new CRISPR-Cas system applications? Also, could the authors specify if these systems were complete? Since detected Cas systems were often fragmented in MAGs and may not be with complete functions.

L378-379: Counting the number of Cas proteins may also need to consider different types of CRISPR-Cas systems. For example, the type V CRISPR-Cas system may have only one effector protein, but the type I CRISPR-Cas systems may have > 10 Cas proteins. Thus comparing Cas protein counts between different phyla may induce > 10 fold over/underestimation if not taking different types of CRISPR-Cas systems in to consideration.

Line 657 to 659: Genomes of the DPANN or CPR superphylum usually lack some markers which are widely present in other archaea or bacteria. Thus, it would be better to use specific markers to estimate the completeness and contamination of these genomes².

L680: Have the authors tried to use newer versions of GTDB? Since r207 has updated a completely different archaeal marker set (from 120 proteins to 53 proteins), and may create a major difference in taxonomy.

Line 727 to 728: The reference number of "antiSMASH" should be 61 rather than 59, and the reference number "92" may be "93"? Please recheck all reference numbers.

references

1. Hedlund, B. P. et al. SeqCode: a nomenclatural code for prokaryotes described from sequence data. *Nat. Microbiol.* 7, 1702–1708 (2022).

2. He, C. et al. Genome-resolved metagenomics reveals site-specific diversity of episymbiotic CPR bacteria and DPANN archaea in groundwater ecosystems. *Nat. Microbiol.* 6, 354–365 (2021).

RE: 2023-08-18. R1

Dear Reviewers,

We sincerely appreciate the time and effort you dedicated to reviewing our manuscript. Your insightful feedback has been instrumental in enhancing the quality and rigor of our work. In response to your valuable comments, we have undertaken revisions that we believe address the concerns raised.

To facilitate your review, we have included a detailed point-by-point response to the feedback and also provided an updated version of the manuscript with all modifications highlighted.

Thank you once again for your constructive guidance and dedication to the peer review process.

Warm regards,

Bin Ma

Bin Ma

Response to the reviewers

Reviewer #1 (Remarks to the Author):

The authors put together a global database of soil metagenomes and assemble thousands of genomes to characterize the functional profiles of soil prokaryotes. The study represents a new catalogue of soil MAGs and their functional characteristics.

I believe this study has a huge amount of work and deserve publication. However, I also found some significant limitations and details that need to be addressed before publication:

RESPONSE: We are deeply appreciative of your positive recognition of our work, particularly in terms of the extensive efforts put forth to construct a comprehensive global database of soil metagenomes and in assembling thousands of genomes to highlight the functional profiles of soil dark matter. And we are gratified that you see value in our new catalogue of soil metagenome-assembled genomes (MAGs) and their functional characteristics.

Your affirmation validates the potential significance and far-reaching implications of our work, as we strive to provide valuable resources for further research in this vital area. However, we also have taken note of your reservations about certain aspects of our study.

Your constructive feedback is indeed beneficial, and we have addressed these points to further refine our study and enhance its potential impact. We understand that addressing these concerns is key to ensuring the quality of our work prior to publication.

Thank you once again for your valuable input.

1. **There is a growing number of soil MAGs catalogues being published in the literature. The study claims that this is “the first excavation of soil microbial dark matter”. However, a recent study published in Nature also included a global characterization of soil MAGs (and those of other habitats) (<https://www.nature.com/articles/s41586-021-04233-4>). What is your study adding to these previous studies?**

RESPONSE: Thank you for bringing up this important point. We acknowledge the valuable contributions of recent studies including the one referenced, which presents a global characterization of the biogeography of prokaryotic genes and also included part of soil MAGs.

While our study may not be the first to explore soil microbial dark matter, **it is distinct in its exclusive focus on soil microbiomes on a global scale. This specificity allowed**

us to undertake an in-depth analysis of this particular niche, expanding the knowledge base on soil microbial diversity.

We acknowledge the contributions of earlier research, particularly those mentioned. Nevertheless, our study differentiates and adds value to the field in several significant ways:

Firstly, our study offers a substantial expansion in scale and relevance to soil metagenomes.

We generated an unprecedented 40,039 MAGs exclusively from 3,304 globally sampled soil metagenome collection, a significant leap from the 3,540 soil bins (including only 416 of high quality and 921 of medium quality) from 243 soil samples reported in the referenced study¹ (See Response Fig. 1) and 2,461 soil MAGs from 1,539 soil samples in the Genomes of Earth's Microbiomes (GEM) catalog² (See L.153 Fig 1d).

The two studies only used one binning tool MetaBAT³, while we use metaWRAP⁴ including three different tools: Metabat³ (v2.12.1), MaxBin⁵ (v2.2.6), CONCOCT⁶ (v1.0.0), which could leverage the strengths of a variety of software to extract and refine high-quality bins from metagenomic data^{4,7}.

Secondly, the geographic distribution of the soil metagenomes in our study significantly expanded from previous works: the soil samples in the earlier studies were primarily sourced from North America and Europe (See Response Fig. 2 and 3). In contrast, our samples provide a much-needed coverage of soil samples from South America, Africa, and Asia (See Response Fig. 4).

Therefore, we have not only substantially increased the number of soil MAGs and soil metagenomes but have also offered a more comprehensive, in-depth, and globally representative investigation of soil microbiomes.

Besides, from the focus of the study, the reference (<https://www.nature.com/articles/s41586-021-04233-4>) you referred to focused on the biogeography and ecology of genes across the global biosphere. **However, our study focused on a more extensive exploration of unknown species-level genome bins (uSGBs), intra-species genome diversity, biosynthetic gene clusters (BGCs), CRISPR-Cas proteins, and novel viral-host associations, which added the contributing novel insights into soil microbial dark matter in the exploration of genetic resources.** Also, the GEM catalog also did the BGCs and virus-host association but lacked the systematic mining of genetic resources, such as the intra-species genome diversity (pangenome construction and SNV CATALOG) and CRISPR-Cas genes.

In summary, our study represents a major leap forward in understanding soil microbial biodiversity from a global scale and unlocking the potential of soil microbial dark matter as a novel source of genetic resources.

Response Fig. 1: Comparison of samples and MAGs number among the current study and other two studies^{1,2}. (HQ: High Quality >90% completeness & <5% contamination; MQ: Medium Quality >50% completeness & <10% contamination).

1. Coelho, L. P. *et al.* Towards the biogeography of prokaryotic genes. *Nature* **601**, 252–256 (2022).
2. Nayfach, S. *et al.* A genomic catalog of Earth's microbiomes. *Nat Biotechnol* **39**, 499–509 (2021).

Response Fig. 2: Global Microbial Gene Catalogue, version 1.

1. Coelho, L. P. *et al.* Towards the biogeography of prokaryotic genes. *Nature* **601**, 252–256 (2022).

Response Fig. 3: A genomic catalog of Earth's microbiomes.

1. Nayfach, S. *et al.* A genomic catalog of Earth's microbiomes. *Nat Biotechnol* **39**, 499–509 (2021).

Response Fig. 4: Recovery of genomes from globally distributed soil metagenomes (Current study).

- Kang, D. D. *et al.* MetaBAT 2: an adaptive binning algorithm for robust and efficient genome reconstruction from metagenome assemblies. *PeerJ* **7**, e7359 (2019).
- Uritskiy, G. V., DiRuggiero, J. & Taylor, J. MetaWRAP—a flexible pipeline for genome-resolved metagenomic data analysis. *Microbiome* **6**, 158 (2018).
- Wu, Y.-W., Simmons, B. A. & Singer, S. W. MaxBin 2.0: an automated binning algorithm to recover genomes from multiple metagenomic datasets. *Bioinformatics* **32**, 605–607 (2016).
- Alneberg, J. *et al.* Binning metagenomic contigs by coverage and composition. *Nat Methods* **11**, 1144–1146 (2014).
- Jia, L. *et al.* A survey on computational strategies for genome-resolved gut metagenomics. *Briefings in Bioinformatics* **24**, bbad162 (2023).

2. In the abstract, it is highlighted that 40,039 MAGs were assembled. Yet, only 3641 supported enough quality (>90% completeness and <5% contamination) to be explored. I think that all functional analyses need to focus on these 3641 MAGs, as the other MAGs are not complete enough to provide reliable information. It is not clear to me if functional analyses were done with all 40,039 MAGs, or only with those showing enough quality. Also, how many of these MAGs are non-redundant?

RESPONSE: Thank you for your insightful comments.

You are absolutely correct, out of the 40,039 assembled metagenome-assembled genomes (MAGs), only 3,641 were deemed high-quality with 90% completeness, less than 5% contamination, and possess the 23S, 16S, and 5S rRNA genes and at least 18 tRNAs according to the Minimum Information about a Metagenome-Assembled Genome (MIMAG) guideline⁸ (See L.153 Fig. 1a). As a result, their functional predictions are expected to be significantly more reliable and accurate but would risk losing if we solely focused on the high-quality MAGs.

Originally, we did the functional analysis for the non-redundant set of 21,077 MAGs, which included 5,184 MAGs (>90% completeness, <5% contamination) and other 15,893 medium-quality MAGs. However, as suggested by your constructive feedback, we also did the functional analysis to focus on the 5,184 representative high-quality MAGs to ensure a comprehensive and more reliable understanding of the soil microbiome functionality (See **Response Fig. 5**) and we have made the change in the manuscript (See **L.249-251 and L.280**). While this subset exhibited a reduced functional diversity in comparison to the entire set of representative 21,077 MAGs (See **Response Fig. 6**), it also distinctly accentuated the significant breadth and expansion of uSGBs in the functional landscape. All in all, both of the results illustrated the uSGBs expanded the functional landscape.

Response Fig. 5: Functional category enrichment differential distribution of 5,184 representative MAGs with 90% completeness, less than 5% contamination.

Response Fig. 6: Functional category enrichment differential distribution of 21,077 representative MAGs.

As for the non-redundant MAGs of our study, we set a threshold of 95% Average Nucleotide Identity (ANI) to delineate redundancy, **which determined 21,077 MAGs to be non-redundant**, forming the core of our data for downstream analyses.

Your feedback is greatly appreciated and we have made it a point to clarify this in our revised manuscript, which brought us to light an important consideration. Thank you again for your attention to detail, which aided us in enhancing the robustness and clarity of our work.

8. Bowers, R. M. *et al.* Minimum information about a single amplified genome (MISAG) and a metagenome-assembled genome (MIMAG) of bacteria and archaea. *nature biotechnology* **35**, 9 (2017).
3. **More information on the precedence of these soil metagenomes needs to be provided in a Supplementary Data. Are these samples from topsoil? Lines 628-633 are difficult to follow. What are in-house samples? Where are those samples coming from? Ecosystems from China?**

RESPONSE: We apologize for any confusion our initial presentation of the metagenomes' source may have caused.

Soil depth was not within the scope of the scientific questions addressed in our study, thus we did not incorporate any selection based on soil depth. Of course, we have also taken your constructive feedback into consideration, and we have made these details clear in the **Supplementary Data 1** of our revised manuscript to make the sample information more informative. Please note that the majority of our soil samples are indeed derived from topsoil, while a portion of them have depth information that is not applicable.

We trust this addition offers the necessary clarification and aids comprehension. Concerning the term "**in-house samples**", we understand the confusion and appreciate your patience. To clarify, we divided all soil metagenomic samples into public and in-house samples. **The in-house samples are the 363 soil samples that our team collected across various regions in China (348) and Europe (15) between 2018-2020, (See L892-896: Five-point sampling method (non-probability sampling was performed in house samples. All soil samples were kept cool using dry ice until visible roots and stones were removed. And then all clean soils were stored at -80°C until DNA extraction)** which were deposited in NCBI under Bioproject PRJNA983538. As for the ecosystems of the in-house samples, detailed information can be found in Supplementary Data 1.

We have elaborated on this in the revised manuscript (**please see L.892-896**). Thank you once again for your attention to detail, as it helps us to improve the clarity of our research work.

L892-896 changed to: In-house samples from this study were sampled by our team across China (348) and Europe (15) in 2018-2020 using a standard sampling protocol⁶⁵. Five-point sampling method (non-probability sampling) was performed in house samples. All soil samples were kept cool using dry ice until visible roots and stones were removed. And then all clean soils were stored at -80°C until DNA extraction.

4. **The title of the paper and the abstract could highlight that this study includes a new catalogue of soil MAGs and their functional profiles.**

RESPONSE: We appreciate your suggestion to highlight the creation of our new MAG catalogue more prominently in the abstract (**See L.25-37**) and title (**See L.1**). We have implemented this revision to better reflect our study's contributions.

L.1 The title changed to: "A Genomic Catalogue of Soil microbiomes Boosts Mining of Biodiversity and Genetic Resources"

We sincerely appreciate your valuable suggestions for the abstract and title.

L.1 changed to: A Genomic Catalogue of Soil microbiomes Boosts Mining of Biodiversity and Genetic Resources

L.25-26 changed to: Soil harbors a vast expanse of unidentified microbes, termed as microbial dark matter, presenting an untapped reservoir of microbial biodiversity and genetic resources, but has yet to be fully explored.

L.31-33 changed to: We also illustrate the pivotal role of uSGBs in augmenting soil microbiome's functional landscape and intra-species genome diversity.

L.35-37 changed to: Our results propose the SMAG catalogue, a novel and expansive genomic resource that brings the soil microbial biodiversity and novel genetic resources to light.

Reviewer #2 (Remarks to the Author):

Comments to the Author

In the manuscript “Soil microbial dark matter explored from genome-resolved metagenomics”, Ma et al. have reconstructed a huge number of metagenome-assembled genomes from about 3000 soil metagenomes with a large proportion of unknown species-level genome bins. This study has expanded archaeal and bacterial genome diversity across the tree of life and enlarged the genomic database, which is very useful for the further study of genetic resources. I appreciate that authors have performed intensive sampling and bioinformatic analyses. I have some comments and suggestions for authors which help to revise the manuscript.

RESPONSE: We are deeply grateful for your constructive feedback on our manuscript "Soil microbial dark matter explored from genome-resolved metagenomics".

We are delighted that you appreciate the significance of our work, specifically the value of the expanded archaeal and bacterial genome diversity and the extensive genomic database we have assembled. We agree that these resources will be instrumental in advancing genetic resources research.

We highly appreciate your thoughtful comments and suggestions which, without a doubt, have assisted us in improving our manuscript.

We thank you again for your input which is crucial in enhancing the quality and depth of our research.

1. **L100: should list ref 25 here as well.**

RESPONSE: We appreciate your suggestion and have added reference at line 124 in the revised manuscript (See L.126), and we have made a check on all references.

L.125-126 changed to: Moreover, 5,184 (13%) of MAGs had completeness $\geq 90\%$ and contamination $< 5\%$, but the absence of all rRNA genes or less than 18 tRNAs²⁴

2. **L151: “kSGBs” should be described when it first appears (L135?).**

RESPONSE: We appreciate your attention to detail. We have provided a clear definition of "kSGBs" (known SGBs) when it is first mentioned in the manuscript (See L.180) to avoid any confusion for the readers.

We appreciate your feedback and hope that these changes improve the clarity and accuracy of our manuscript.

L.179-180 changed to: The proportion of singleton MAGs in uSGBs (71.2%) was substantially higher than in **known SGBs (kSGBs)** (50.0%) (Fig. 2a), indicating the

critical contribution of the SMAG catalogue in recovering rare species of soil microbiomes.

- 3. L184: the authors may consider proposing new taxonomic levels based on GTDB and SeqCode1 rather than 16S rRNA. Also, "16s" should be "16S".**

RESPONSE: We can't agree more with your proposal, and we clarified it in the manuscript (See L.223-229). We have acknowledged that 16S rRNA may not capture the full microbial diversity. The 16S rRNA gene is a widely used molecular marker for taxonomic identification and phylogenetic analysis of microorganisms. However, relying solely on 16S rRNA gene-based classification for metagenome-assembled genomes (MAGs) and single-amplified genomes (SAGs) is gradually becoming inadequate to meet the demands of classifying the ever-increasing number of uncultured microorganisms due to fragmentation during assembly and leading to its low coverage. **However, the taxonomy of GTDB (Genome Taxonomy Database)⁹ and SeqCode¹⁰ proposed a promising solution for the reasonable classification of the vast number of MAGs or SAGs.**

Through the annotation provided by GTDB, we have enthusiastically adopted it in the study (See **Supplementary Data 2**), which achieved a unified and precise taxonomy and had the potential to be widely accepted and utilized by researchers across various fields.

Also, SeqCode enables valid publication of names of prokaryotes based upon isolate genome, metagenome-assembled genome or single-amplified genome sequences, which provides a reproducible and objective framework for the nomenclature of all prokaryotes. However, to further elaborate, assigning names to the numerous unannotated species-level genome bins (uSGBs) in the SMAG catalogue using SeqCode is a significant, time-consuming task so we have adopted the GTDB taxonomy in the study.

Additionally, we would like to express our appreciation to for your thoughtful suggestion to employ SeqCode for nomenclature. The insight will undeniably enhance the quality of our research and future endeavors.

All in all, embracing these advanced taxonomic tools has enriched our understanding of microbial diversity and has contributed significantly to the advancement of microbial taxonomy and evolutionary studies in the context of deep excavation research.

Also, we have corrected "16s" to "16S" throughout the text.

Once again, we extend our gratitude for your valuable advice.

L.223-229 changed to: Two bacterial SGBs were potentially unannotated phylum-level genome bins (uPGBs) with completeness and contamination at 90.65%-2.44%, and 90.96%-1.10%, respectively, which indeed illustrated the underestimated diversity

of the soil microbial dark matter and highlighted the pressing need for continued exploration of the soil microbiome.

9. Chaumeil, P.-A., Mussig, A. J., Hugenholtz, P. & Parks, D. H. GTDB-Tk v2: memory friendly classification with the genome taxonomy database. *Bioinformatics* **38**, 5315–5316 (2022).

10. Hedlund, B. P. *et al.* SeqCode: a nomenclatural code for prokaryotes described from sequence data. *Nat Microbiol* **7**, 1702–1708 (2022).

4. Line 206 to 209: Due to the large number of uSGBs, the higher gene number in uSGBs seems predictable. The proportion of each COG functional category may be suited to studying the function difference between uSGBs and kSGBs.

RESPONSE: We appreciate your suggestion regarding the exploration of functional differences between uSGBs and kSGBs based on the proportion of each COG functional category.

However, it is important to note that the higher gene number in uSGBs does not necessarily indicate a distinct functional profile from kSGBs. This functional diversity is likely dependent on a complex interplay of many factors beyond the simple numerical abundance of genes.

To address this point, we included a comprehensive analysis comparing the proportion of each COG functional category between uSGBs and kSGBs (**See Response Fig. 8 and Response Table 1**). This analysis illustrated there is little difference in COG functional category between uSGBs and kSGBs. However, about 19.4% COG functional category was annotated as function unknown (**See Response Fig. 7 and Response Table 1**).

In conclusion, your suggestion greatly enhanced our analysis, and we believe the added comparison provides further insights into the functional capacities of these previously unexplored soil microbes.

We appreciate your suggestion regarding the exploration of functional differences between uSGBs and kSGBs based on the proportion of each COG functional category.

However, it is important to note that the higher gene number in uSGBs does not necessarily indicate a distinct functional profile from kSGBs. This functional diversity is likely dependent on a complex interplay of many factors beyond the simple numerical abundance of genes.

To address this point, we included a comprehensive analysis comparing the proportion of each COG functional category between uSGBs and kSGBs (**See Response Fig. 8 and Response Table 1**). This analysis illustrated there is little difference in COG functional category between uSGBs and kSGBs. However, about 19.4% COG functional category was annotated as function unknown (**See Response Fig. 7 and Response Table 1**).

In conclusion, your suggestion greatly enhanced our analysis, and we believe the added comparison provides further insights into the functional capacities of these previously unexplored soil microbes.

Proportion of COG category from the SMAG

Response Fig. 7: The proportion of COG function category of the SMAG catalog.

COG_category	info	Gene number
L Amino acid transport and metabolism	kSGBs	394497
i Amino acid transport and metabolism	uSGBs	991622
? Carbohydrate transport and metabolism	kSGBs	286654
i Carbohydrate transport and metabolism	uSGBs	702963
j Cell cycle control, cell division, chromosome partitioning	kSGBs	62661
? Cell cycle control, cell division, chromosome partitioning	uSGBs	148406
L Cell motility	kSGBs	72189
i Cell motility	uSGBs	176472
j Cell wall/membrane/envelope biogenesis	kSGBs	333065
i Cell wall/membrane/envelope biogenesis	uSGBs	812394
i Chromatin structure and dynamics	kSGBs	2513
? Chromatin structure and dynamics	uSGBs	6206
? Coenzyme transport and metabolism	kSGBs	202922
L Coenzyme transport and metabolism	uSGBs	488874
j Cytoskeleton	kSGBs	3355
? Cytoskeleton	uSGBs	8945
j Defense mechanisms	kSGBs	89059
i Defense mechanisms	uSGBs	231976
? Energy production and conversion	kSGBs	344684
L Energy production and conversion	uSGBs	872394
L Extracellular structures	kSGBs	1594
i Extracellular structures	uSGBs	3842
? Function unknown	kSGBs	964071
i Function unknown	uSGBs	2327812
j Inorganic ion transport and metabolism	kSGBs	265814
? Inorganic ion transport and metabolism	uSGBs	623085
L Intracellular trafficking, secretion, and vesicular transport	kSGBs	122858
i Intracellular trafficking, secretion, and vesicular transport	uSGBs	305897
i Lipid transport and metabolism	kSGBs	198524
j Lipid transport and metabolism	uSGBs	526708
i Nuclear structure	kSGBs	3
? Nuclear structure	uSGBs	2
? Nucleotide transport and metabolism	kSGBs	125098
L Nucleotide transport and metabolism	uSGBs	293541
j Posttranslational modification, protein turnover, chaperones	kSGBs	182890
? Posttranslational modification, protein turnover, chaperones	uSGBs	459938
i Replication, recombination and repair	kSGBs	255038
j Replication, recombination and repair	uSGBs	579874
i RNA processing and modification	kSGBs	1301
? RNA processing and modification	uSGBs	3102
L Secondary metabolites biosynthesis, transport and catabolism	kSGBs	143255
i Secondary metabolites biosynthesis, transport and catabolism	uSGBs	404446
? Signal transduction mechanisms	kSGBs	260303
i Signal transduction mechanisms	uSGBs	676359
j Transcription	kSGBs	337809
? Transcription	uSGBs	842811
L Translation, ribosomal structure and biogenesis	kSGBs	259347
i Translation, ribosomal structure and biogenesis	uSGBs	603329

Response Table 1: The statistics of gene number of COG function category.

Response Fig. 8: The proportion of COG category genes between kSGB and uSGB.

5. L214: Could the authors clarify the clustering method? Why chose 90% amino acid identity as the threshold?

RESPONSE: Thank you for your question regarding our choice of clustering method and the 90% amino acid identity threshold for the pangenome analysis. I'm glad to provide some clarification on these points.

First, the clustering method we used is a part of Roary¹¹ v3.12.0, a high-speed stand-alone pan-genome pipeline, which can take annotated assemblies in GFF3 format and calculate the pan-genome. Roary is not only known for its speed and efficiency in large-scale data but also for its capacity to reduce errors in clustering, thereby providing a more reliable analysis^{12,13}.

Second, the choice of 90% amino acid identity as the threshold is based on a number of factors that are typically considered in pan-genome analyses. Here are a few key reasons:

1) At 90% amino acid identity, this level of identity is often used **as a threshold to define a 'core' genome** – genes that are shared by all individuals in a species and are typically essential for basic cellular functions^{14,15}.

2) The 90% threshold is a compromise that **balances sensitivity and specificity**. Both too high an identity threshold (e.g., 95% or 99%) and a lower threshold (e.g., 85% or 80%) would affect the sensitivity and specificity.

3) The choice of a 90% identity threshold is fairly standard in pan-genome analyses and comparative genomics, **which is also adopted in many associated researches¹⁶⁻¹⁸**.

(<https://www.nature.com/articles/s41587-020-0603-3>,

<https://www.nature.com/articles/s41586-023-06173-7>,

<https://www.nature.com/articles/s41586-021-04332-2>).

By using this threshold, the results of the study can be more easily compared to other studies in the field.

Your feedback is appreciated, and we acknowledge that the choice of threshold can greatly influence the results of the clustering and subsequent pan-genome analysis. Therefore, we have further clarified the reasoning behind this choice in the revised manuscript.

12. Scholz, M. *et al.* Large scale genome reconstructions illuminate Wolbachia evolution. *Nat Commun* **11**, 5235 (2020).

13. Vassallo, C. N., Doering, C. R., Littlehale, M. L., Teodoro, G. I. C. & Laub, M. T. A functional selection reveals previously undetected anti-phage defence systems in the *E. coli* pangenome. *Nat Microbiol* **7**, 1568–1579 (2022).

14. Steinegger, M. & Söding, J. Clustering huge protein sequence sets in linear time. *Nat Commun* **9**, 2542 (2018).

15. Franzosa, E. A. *et al.* Species-level functional profiling of metagenomes and metatranscriptomes. *Nat Methods* **15**, 962–968 (2018).

16. Almeida, A. *et al.* A unified catalog of 204,938 reference genomes from the human gut microbiome. *Nat Biotechnol* (2020) doi:10.1038/s41587-020-0603-3.

17. Gao, Y. *et al.* A pangenome reference of 36 Chinese populations. *Nature* **619**, 112–121 (2023).

18. Edgar, R. C. *et al.* Petabase-scale sequence alignment catalyses viral discovery. *Nature* **602**, 142–147 (2022).

6. **Supplementary Figure. 5d: “Number of cas protein” should be “Proportion of cas protein”.**

RESPONSE: Thank you for catching this error. We have corrected the caption of the Supplementary Figure 5d to say “Proportion of cas protein” in the revised manuscript. (See L.1361).

L.1361 Supplementary Figure 5d changed to:

7. **L266-267: Have the authors considered using adding dN/dS ratio rather than SNV alone as proxies for connecting genetic diversity to ecosystem functions? Because synonymous mutations are neutral and do not contribute to changes in protein functions. Thus, this part of SNV (synonymous) may not lead to niche breath changes and adaptability to different environments for certain microorganisms.**

RESPONSE: We deeply appreciate your insightful suggestions.

Indeed, synonymous mutations, which don't modify protein sequences, are traditionally considered neutral or nearly so. Nonetheless, the role of such mutations is under active discussion in the scientific community. Increasing evidence suggests that synonymous mutations can impact gene expression, protein folding, and fitness and drive adaptive evolution, which suggests that this class of mutation may be underappreciated as a cause of adaptation and evolutionary dynamics^{19–21}. Recent work even found that most synonymous mutations are strongly non-neutral²². Therefore, we originally utilize all types of nucleotide variations to gain a more comprehensive understanding of the view of genetic variations of soil microorganisms.

Also, we agreed that minimizing the influence of synonymous mutations can yield richer insights. We additionally constructed a dataset that filters out synonymous mutations, conducting subsequent analyses of filtered SNVs (exclude non-synonymous SNVs) between uSGB and kSGB, also the non-synonymous SNVs across different phyla in the result (See Fig. 3h L.280 and L.339-367).

That being said, the merit of using the dN/dS ratio as a more direct measure of evolutionary pressure cannot be understated. It provides a clearer distinction between the influences of synonymous and non-synonymous mutations on genetic diversity and adaptability. Given your feedback, we have re-evaluated our approach and incorporated analyses using the ratio of non-synonymous to synonymous polymorphism (pN/pS)²³ to offer a more comprehensive view (See L.339-367 and L.1011-1014). And we observed variations in SNV density and pN/pS ratios across different phyla which illustrate the diverse niche widths of these species and their varying capacities to acquire and allocate soil resources.

Once again, thank you for your valuable feedback, which has significantly enriched our manuscript.

L.1011-1014 changed to: To filter the synonymous SNVs, we calculated the synonymous ratio with the house script `snv-filter.py`, and we estimated the ratio of non-synonymous to synonymous polymorphism rates⁴⁴ (pN/pS) to evaluate the genetic diversity.

L.339-367 changed to: Notably, we observed a divergence in the density of filtered-SNVs between kSGBs and uSGBs across most dominant phyla (Fig. 3g, Supplementary Figure 3f). Additionally, a majority of the phyla exhibited relatively low pN/pS ratios (pN/pS <1) (Fig. 3h and Supplementary Data 3). This suggests that the evolution of soil microbial organisms might be more influenced by long-term purifying selection and drift, rather than by rapid adaptations to specific environments¹. While species from Patescibacteria possess the smallest genome sizes, displayed the lowest ns-SNV density coupled with the highest pN/pS ratios, possibly owing to their reduced redundant and non-essential functions that enable them to maintain community stability³³. These findings suggest that the SMAG catalogue encompasses a significant amount of intraspecific SNVs. The observed variations in SNV density and pN/pS ratios across different phyla underscore the diverse niche widths of these species and their varying capacities to acquire and allocate soil resources⁴³.

L.280 Fig. 3f changed to:

L.280 Fig. 3g changed to:

L.280 added Fig. 3h:

L.599-601 changed to: And the divergent pN/pS ratios indicated the soil microbiome experienced a strong purifying selection, which may highlight the environmental adaptability of species within the community, emphasizing a balance where deleterious genetic variations are minimized⁴⁵.

L.1338 Supplementary Figure 3f changed to:

19. Sharon, E. *et al.* Functional Genetic Variants Revealed by Massively Parallel Precise Genome Editing. *Cell* **175**, 544-557.e16 (2018).
20. She, R. & Jarosz, D. F. Mapping Causal Variants with Single-Nucleotide Resolution Reveals Biochemical Drivers of Phenotypic Change. *Cell* **172**, 478-490.e15 (2018).
21. Bailey, S. F., Hinz, A. & Kassen, R. Adaptive synonymous mutations in an experimentally evolved *Pseudomonas fluorescens* population. *Nat Commun* **5**, 4076 (2014).
22. Shen, X., Song, S., Li, C. & Zhang, J. Synonymous mutations in representative yeast genes are mostly strongly non-neutral. *Nature* **606**, 725–731 (2022).
23. Jeffares, D. C., Tomiczek, B., Sojo, V. & dos Reis, M. A beginners guide to estimating the non-synonymous to synonymous rate ratio of all protein-coding genes in a genome. *Methods Mol Biol* **1201**, 65–90 (2015).

8. L319-321: Are there any results support that the identified BGCs in this study were novel compared to existing studies? Since the authors suggest that BGCs found in this study may lead to the development of new drugs and therapeutics.

RESPONSE: Thank you for your insightful question. In this study, we indeed identified several biosynthetic gene clusters (BGCs), which show significant potential for the discovery of new therapeutic compounds.

GEM also predicted 104,211 from 52,515 MAGs from multi-environments without doing GCFs clustering, and a study on the biosynthetic potential of the global ocean microbiome²⁴ predicted a total of 39,055 BGCs and clustered them into 6,907 non-redundant gene cluster families (GCFs), while Wei et al.²⁵ predicted 70,011 from 24,536 marine MAGs, which both of them focus on the ocean biosynthetic potential. However, our study did the first global scale soil biosynthetic potential analysis, and we identified 70,081 putative BGCs from the **non-redundant representative 21,077 MAGs**, of which 43,169 BGCs with a length ≥ 5 kb and cluster them into 33,941 GCFs (See Supplementary Data 4 and Response Fig.7), which suggested that our study not

only identified a greater number of BGCs but also revealed a larger diversity of GCFs than previous studies, highlighting the unique contribution of our soil microbial potentiality to the discovery of potential new drug resources.

Response Fig. 7: Comparison of BGCs and GCFs identified from MAGs among current study and other three recent studies.

Secondly, given that the microbial diversity in soil is vast and largely unexplored, it is reasonable to assume that novel BGCs can be uncovered. These BGCs, found in novel or underrepresented microbial taxa, may harbor the potential for new drug and therapeutic discovery.

In the future, more focused investigations, including gene synthesis, heterologous expression, and compound isolation, are needed to validate the potential of these BGCs. We hope that our study has provided a valuable resource for future explorations in the quest for novel drugs and therapeutics.

24. Paoli, L. *et al.* Biosynthetic potential of the global ocean microbiome. *Nature* 1–8 (2022) doi:10.1038/s41586-022-04862-3.

25. Wei, B. *et al.* Global analysis of the biosynthetic chemical space of marine prokaryotes. *Microbiome* 11, 144 (2023).

9. L364: “cas” should be capitalized as “Cas” throughout the text.

RESPONSE: Thank you for pointing this out. We have corrected this and ensured that the term is capitalized as "Cas" throughout the manuscript.

10. L369: What’s before “of which”?

RESPONSE: We apologize for the confusion caused by the incomplete sentence. The phrase "of which" was intended to introduce 245 MAGs of 563 MAGs that possessed less than 10 Cas-associated genes (Fig. 5d) (See L.489-491). We have reviewed this

section to ensure the sentence now provides clear and complete information. Thank you for bringing this to our attention.

L.489-491 changed to: 245 MAGs (43.5%) possessed less than 10 Cas-associated genes (Fig. 5d) and only 1,611 Cas-associated genes (18.8%) were identified with certain Cas-associated genes (Fig. 5g, Supplementary Figure 5d).

11. L372-374: similar to BCGs, what specific results indicate that the finding in this study may facilitate the development of new CRISPR-Cas system applications? Also, could the authors specify if these systems were complete? Since detected Cas systems were often fragmented in MAGs and may not be with complete functions.

RESPONSE: Thank you for your insightful questions. You correctly pointed out the potential for our findings to be applied in the development of new CRISPR-Cas system applications. The diversity of Cas proteins identified in our study expands the current understanding of CRISPR-Cas systems, providing novel candidates for gene-editing tool development. Each Cas protein variant potentially presents unique properties that may have practical value, such as differing precision, efficiency, or suitability for certain cell types. By the way, we have done some work unpublished on the CRISPR-Cas system transformation and application, which showed that the two new Cas13gs identified from our SMAG catalogue showed comparative endogenous mRNA knockdown efficiency compared with other efficient Cas13 proteins.

However, these results are currently undergoing, so we apologize for not being able to include these data in the Supplementary Figure at this stage.

As for the completeness of the identified CRISPR-Cas systems, we acknowledge that in metagenome-assembled genomes (MAGs), **these systems can often be fragmented**, potentially limiting their functionality.

In our analyses, we aimed to mine novel Cas proteins for powerful genome editing tools. Still, it's essential to emphasize that **experimental validation would be necessary to confirm the functionality** of these putatively identified systems. Also, similar works have been done on the modification and deep excavation of CRISPR-Cas systems from genomes²⁶⁻²⁸. In future work, we will further validate these systems and explore their potential practical applications in more depth.

Your comments have highlighted an important aspect of our study and we appreciate your helpful suggestions. They have helped us clarify the significance and potential applications of our findings.

26. Burstein, D. *et al.* New CRISPR–Cas systems from uncultivated microbes. *Nature* **542**, 237–241 (2017).

27. Gao, L. *et al.* Diverse enzymatic activities mediate antiviral immunity in prokaryotes. *Science* **369**, 1077–1084 (2020).

28. Doron, S. *et al.* Systematic discovery of antiphage defense systems in the microbial pangenome. *Science* **359**, eaar4120 (2018).

12. L378-379: Counting the number of Cas proteins may also need to consider different types of CRISPR-Cas systems. For example, the type V CRISPR-Cas system may have only one effector protein, but the type I CRISPR-Cas systems may have > 10 Cas proteins. Thus comparing Cas protein counts between different phyla may induce > 10 fold over/underestimation if not taking different types of CRISPR-Cas systems in to consideration.

RESPONSE: Thank you for raising an important point regarding the complexity and diversity of CRISPR-Cas systems. We agree that different CRISPR-Cas systems may consist of a varying number of Cas proteins, and this should be considered when analyzing the presence and distribution of these systems across different phyla.

However, the rapid evolution of most cas genes²⁹ caused a great challenge for the consistent annotation of Cas proteins and for the subsequent classification of CRISPR–Cas systems²⁸. Besides, it is difficult to confirm the activity of naturally occurring CRISPR–Cas systems without experimental verification³⁰.

Therefore, our study primarily centered on a simplified enumeration of Cas protein resources (See L.530 and L. 492-494). Our intent was to provide a broad perspective of the potential Cas protein reservoir within metagenomes, which often resulted in fragments, rather than for the complete CRISPR–Cas systems.

We deeply value your insights, and we sincerely appreciate your contribution.

L.488-489 changed to: Notably, we identified 42 Cas 9, which were potentially engineered for powerful genome editing tools⁵⁵.

L.492-494 changed to: The collection of Cas protein family profiles is a resource for the identification of CRISPR–Cas systems³, which also illustrates the necessity and importance of mining the soil microbiome.

L.527-529 changed to: uSGBs offered a great deal of unknown information about Cas proteins from the soil microbiome. This also demonstrates the utility of metagenomic mining for gene editing tools development.

29. Ks, M. *et al.* Evolution and classification of the CRISPR-Cas systems. *Nature reviews. Microbiology* **9**, (2011).

30. Makarova, K. S. *et al.* An updated evolutionary classification of CRISPR–Cas systems. *Nat Rev Microbiol* **13**, 722–736 (2015).

13. Line 657 to 659: Genomes of the DPANN or CPR superphylum usually lack some markers which are widely present in other archaea or bacteria. Thus, it would be better to use specific markers to estimate the completeness and contamination of these genomes².

RESPONSE: We appreciate and concur with the insightful comments about the challenges in assessing the completeness and contamination of the genomes belonging to DPANN or Candidate Phyla Radiation (CPR) superphyla using traditional methods. This is primarily because these genomes often lack some marker genes that are ubiquitous in other archaea or bacteria.

CheckM³¹ has been employed extensively to predict the completeness and contamination of metagenome-assembled genomes (MAGs) through lineage-specific marker genes. However, its limitation surfaces when there are no 'universal' marker genes available, resulting in reduced accuracy and sensitivity. This issue is particularly noticeable in the case of CPR (Patescibacteria) and DPANN, which are known to lack certain markers.

In our dataset of 40,039 MAGs, we have 1,865 CPR MAGs and 341 DPANN MAGs. These MAGs have predicted completeness greater than 50% and contamination less than 10% by CheckM (Please see Supplementary Data 2). However, given this limitation of CheckM, we adopted CheckM2³² to reevaluate these genomes. CheckM2, a machine learning-based tool, allows us to predict the completeness and contamination of bacterial and archaeal genomes without relying on predefined lineage-specific marker sets. Interestingly, our results revealed that **only about 0.16% of CPR MAGs (34 MAGs) and approximately 0.02% of DPANN MAGs (4 MAGs) among the representative 21,077 soil metagenome-assembled genomes (SMAGs) failed to meet the criterion of >50% completeness and <10% contamination. This small percentage would not impact our central conclusion that the catalogue of SMAG provided valuable insight into the biodiversity and novel genetic resources of soil microbiota. **Also, we have also included the results from CheckM2 in Supplementary Data 2.****

We appreciate your suggestion about using specific markers to assess genome completeness and contamination, and we believe that our approach aligns with your suggestion can provide a comprehensive assessment of our dataset.

31. Parks, D. H., Imelfort, M., Skennerton, C. T., Hugenholtz, P. & Tyson, G. W. CheckM: assessing the quality of microbial genomes recovered from isolates, single cells, and metagenomes. *Genome Res.* **25**, 1043–1055 (2015).

32. Chklovski, A., Parks, D. H., Woodcroft, B. J. & Tyson, G. W. CheckM2: a rapid, scalable and accurate tool for assessing microbial genome quality using machine learning. *Nat Methods* 1–10 (2023) doi:10.1038/s41592-023-01940-w.

14. L680: Have the authors tried to use newer versions of GTDB? Since r207 has updated a completely different archaeal marker set (from 120 proteins to 53 proteins), and may create a major difference in taxonomy.

RESPONSE: We appreciate your insightful observation regarding the version of the Genome Taxonomy Database (GTDB)⁹ we used in our study. We concur that it's vital to stay abreast with the latest versions of databases in genomic research to ensure the most accurate and up-to-date results.

In the context of our work, we initially utilized GTDB r202 for the taxonomy assignment. Nonetheless, we acknowledge the modifications incorporated in the newer version, GTDB r207/r214, which has substantially updated the archaeal marker set.

To verify the robustness of our results, we undertook a comparative analysis between the two versions focusing on the primary bacterial and archaeal taxa (**See Response Fig. 9**). Interestingly, our findings revealed minor differences, which do not impact our overall conclusions or the integrity of the Soil Metagenome-Assembled Genomes (SMAG) catalogue we presented. **Besides, we have updated the new version of GTDB release 214 taxonomic annotation in the SAMG catalogue interface web (<https://smag.microbmalab.cn>).**

While our study's central objective is the construction of a comprehensive SMAG catalogue, serving as a valuable resource for uncovering the 'soil microbial dark matter' and extracting novel genetic resources, we understand the pivotal role of taxonomy in the process of novel taxa discovery.

Assuring the scientific community of our commitment to employing the most accurate and updated tools, we plan to leverage the latest version of GTDB in our future research endeavors, especially those concentrating on the identification of new taxa. This approach will ensure we deliver the highest degree of accuracy and maintain real-time relevance.

We thank you for bringing up this important aspect, and we will clearly address this in the methodology section of our revised manuscript to maintain full transparency and enhance the rigour of our research.

Thank you again for your valuable contribution to the improvement of our manuscript.

Response Fig. 9: The comparison of main bacteria and archaea of SMAG catalogue taxonomic annotation by GTDB release 202 and the latest GTDB release 214.

9. Chaumeil, P.-A., Mussig, A. J., Hugenholtz, P. & Parks, D. H. GTDB-Tk v2: memory friendly classification with the genome taxonomy database. *Bioinformatics* **38**, 5315–5316 (2022).

15. Line 727 to 728: The reference number of “antiSMASH” should be 61 rather than 59, and the reference number “92” may be “93”? Please recheck all reference numbers.

RESPONSE: We apologize for the oversight. We have meticulously reviewed all references in the manuscript to ensure their accuracy. (See L.991-992).

L.991-992 changed to: Secondary-metabolite BGCs of SMAG were identified using antiSMASH⁵⁸ (v6.1) with default settings and the corresponding database (v5.0)⁸⁸.

REVIEWERS' COMMENTS

Reviewer #1 (Remarks to the Author):

The authors satisfactorily addressed my previous comments. Please carefully edit the language.

Reviewer #2 (Remarks to the Author):

I appreciate authors on thoroughly revising the manuscript to address my comments and suggests. I believe this study has compiled a global database of soil metagenomes and assembled thousands of genomes to characterize the functional profiles of soil prokaryotes, and provided a new catalog of soil MAGs and their functional characteristics, which should be very useful for relevant research. The job has been done very well, and I have no further questions regarding the revised document.

RE: 2023-10-12. R2

Dear Reviewers,

We extend our deepest gratitude for your thorough review and the positive affirmation on our manuscript. It is truly encouraging to have esteemed experts like you recognize and validate the merit of our work. Such acknowledgment motivates us to further advance our research endeavors.

Your constructive comments have not only helped elevate the quality of our paper but also reaffirmed its significance in the scientific community.

Your dedication to the integrity and rigor of the peer review process is deeply appreciated. we sincerely thank you for your guidance and the trust you've placed in our research.

Best regards,

Bin Ma

REVIEWERS' COMMENTS

Reviewer #1 (Remarks to the Author):

The authors satisfactorily addressed my previous comments. Please carefully edit the language.

RESPONSE: Thank you for recognizing our efforts to address your previous comments. We are committed to further refining the language of our manuscript to ensure clarity and precision. Your constructive feedback has been invaluable in improving the quality of our work, and we are truly appreciative.

Reviewer #2 (Remarks to the Author):

I appreciate authors on thoroughly revising the manuscript to address my comments and suggests. I believe this study has compiled a global database of soil metagenomes and assembled thousands of genomes to characterize the functional profiles of soil prokaryotes, and provided a new catalog of soil MAGs and their functional characteristics, which should be very useful for relevant research. The job has been done very well, and I have no further questions regarding the revised document.

RESPONSE: We are truly grateful for your thoughtful feedback throughout the revision process. Your invaluable comments played a pivotal role in refining and elevating our manuscript to its current state. Recognizing the depth and scope of our study in your response reinforces the importance of the rigorous review process, and we are dedicated to ensuring the quality and relevance of our research for the broader scientific community. We express our sincerest gratitude again for your time and expertise throughout the review process.